# ON UNI-MODAL FEATURE LEARNING IN SUPERVISED MULTI-MODAL LEARNING

## ABSTRACT

We abstract the features of multi-modal data into 1) *uni-modal features*, which can be learned from uni-modal training, and 2) *paired features*, which can *only* be learned from cross-modal interaction. Multi-modal joint training is expected to benefit from cross-modal interaction on the basis of ensuring uni-modal feature learning. However, recent late-fusion training approaches still suffer from insufficient learning of uni-modal features on each modality, and we prove that this phenomenon does hurt the model's generalization ability. Given a multi-modal task, we propose to choose targeted late-fusion learning method from **Uni-M**odal **E**nsemble (UME) and the proposed **Uni-M**odal **T**eacher (UMT), according to the distribution of uni-modal and paired features. We demonstrate that, under a simple guiding strategy, we can achieve comparable results to other complex late-fusion or intermediate-fusion methods on multi-modal datasets, including VGG-Sound, Kinetics-400, UCF101, and ModelNet40.

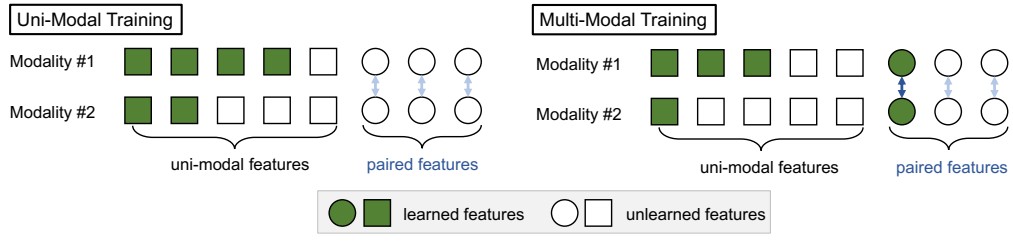

Figure 1: **Overview of Modality Laziness**. Although multi-modal joint training provides the opportunity for cross-modal interaction to learn paired features, the model easily saturates and ignores the uni-modal features that are hard to learn but also important to generalization.

## 1 INTRODUCTION

Multi-modal signals, *e.g.*, vision, sound, text, are ubiquitous in our daily life, allowing us to perceive the world through multiple sensory systems. Inspired by the crucial role that multi-modal interactions play in human perception and decision (Smith & Gasser, 2005), substantial efforts have been made to build effective and reliable computational multi-modal systems in fields like multimedia computing (Wang et al., 2020; Xiao et al., 2020), representation learning (Arandjelovic & Zisserman, 2017; Radford et al., 2021) and robotics (Chen et al., 2020a).

According to how the features of multi-modal data can be learned, we abstract them into two categories: (1) *uni-modal features*, which can be learned from uni-modal training, and (2) *paired features*, which can *only* be learned from cross-modal interaction. In this paper, we focus on multimodal tasks where uni-modal priors are meaningful [1] (Kay et al., 2017; Chen et al., 2020b). Ideally, we hope that multi-modal joint training can learn paired features through cross-modal interactions on the basis of ensuring that enough uni-modal features are learned.

---

[1] Uni-modal prior here means that we get predictions only according to one modality in multi-modal tasks.

However, recent late-fusion methods still suffer from learning insufficient uni-modal representations of each modality (Peng et al., 2022). We term this phenomenon as *Modality Laziness* and illustrate that in Figure 1. We theoretically characterize Modality Laziness and prove that **it does hurt the generalization ability of the model**, especially when uni-modal features are dominant in the given task. Besides the laziness problem, another shortcoming of recent late-fusion approaches is that they are complex to implement. For example, G-Blending (Wang et al., 2020) needs an extra split of data to estimate the overfitting-to-generalization ratio to re-weight the losses and then re-train the model again and again. Peng et al. (2022) proposes OGM-GE, which dynamically adjusts the gradients of different modalities during training. However, it needs to tune too many hyper-parameters [2], including the start and end epoch of the gradient modulation, an "alpha" used to calculate the coefficients for the modulation and whether adaptive Gaussian noise Enhancement (GE) is needed. The more complicated thing is that these hyper-parameters need to be re-tuned on new datasets.

To this end, more simple and effective methods are urgently needed. We pay attention to the learning of uni-modal features and propose to choose targeted late-fusion training method from **U**ni-**M**odal **E**nsemble (UME) and proposed **U**ni-**M**odal **T**eacher (UMT) according to the distribution of uni-modal and paired features. If both uni-modal and paired features are essential, UMT is effective, which helps multi-modal models better learn uni-modal features via uni-modal distillation; if both modalities have strong uni-modal features and paired features are not important enough, UME is properer, which combines predictions of uni-modal models and completely avoids insufficient learning of uni-modal features. We also provide an empirical trick to decide which one to use. Under this guidance, we achieve comparable results to other complex late-fusion or intermediate-fusion methods on multiple multi-modal datasets, including VGG-Sound (Chen et al., 2020b), Kinetics-400 (Kay et al., 2017), UCF101 (Soomro et al., 2012) and ModelNet40 (Wu et al., 2022).

## 2 RELATED WORK

**Multi-modal training approaches** aim to train a multi-modal model by using all modalities simultaneously (Liang et al., 2021), including audio-visual classification (Peng et al., 2022; Xiao et al., 2020; Panda et al., 2021), action recognition (Wang et al., 2020; Panda et al., 2021), visual question answering (Agrawal et al., 2018) and RGB-D segmentation (Park et al., 2017; Hu et al., 2019; Seichter et al., 2020). There are several different fusion methods, including early/middle fusion (Seichter et al., 2020; Nagrani et al., 2021; Wu et al., 2022) and late fusion (Wang et al., 2020; Peng et al., 2022; Fayek & Kumar, 2020). In this paper, we mainly consider the late-fusion methods following Wang et al. (2020), which is convenient and straightforward to evaluate the learning of uni-modal features. We demonstrate that simple late-fusion approaches can outperform approaches with more complex model architecture (Wu et al., 2022; Xiao et al., 2020).

**Multi-modal learning theory.** The research on multi-modal learning theory is still at an early age. A line of work focuses on understanding multi-view tasks (Amini et al., 2009; Xu et al., 2013; Arora et al., 2016; Allen-Zhu & Li, 2020), and our assumption on the data structure partially stems from Allen-Zhu & Li (2020). Huang et al. (2021) explains multi-modal learning is potentially better than uni-modal learning and Huang et al. (2022) explains why failure exists in multi-modal learning. Our paper investigates the different types of features in multi-modal data and provides solutions for the weakness of multi-modal learning.

**Knowledge distillation** was introduced to compress the knowledge from an ensemble into a smaller and faster model but still preserve competitive generalization power (Buciluǎ et al., 2006; Hinton et al., 2015; Tian et al., 2019; Gou et al., 2021; Allen-Zhu & Li, 2020). In this paper, we propose Uni-Modal Teacher to leverage uni-modal distillation for joint training to help the learning of uni-modal features, without involving cross-modal knowledge distillation (Pham et al., 2019; Gupta et al., 2016; Tan & Bansal, 2020; Garcia et al., 2018; Luo et al., 2018).

## 3 ANALYSIS, LEARNING GUIDANCE AND THEORY

In this section, we show the drawbacks and advantages of joint training. On one hand, joint training results in insufficient learning of uni-modal features (Modality Laziness). On the other hand, it

---

[2] https://github.com/GeWu-Lab/OGM-GE_CVPR2022

Table 1: Top 1 test accuracy (in %) of linear evaluation on encoders from various multi-modal late-fusion training methods and uni-modal training on VGG-Sound and UCF101.

| Method | VGG-Sound | | UCF101 | |
|---|---|---|---|---|
| | RGB Encoder | Audio Encoder | RGB Encoder | Opt-Flow Encoder |
| Linear-Fusion | 15.56 | 43.44 | 75.66 | 48.08 |
| MLP-Fusion | 14.52 | 40.01 | 75.65 | 51.89 |
| Attention-Fusion | 13.31 | 43.97 | 74.84 | 7.72 |
| G-Blending | 17.69 | 43.90 | 74.91 | 44.49 |
| OGM-GE | 15.60 | 41.95 | 73.54 | 65.03 |
| Uni-Modal Training | **23.17** | **45.15** | **77.08** | **74.99** |

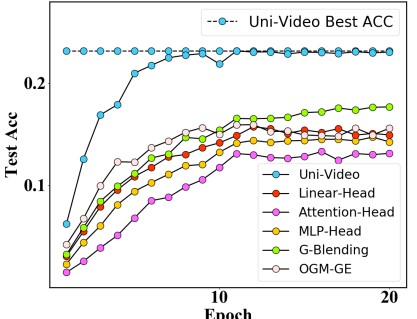 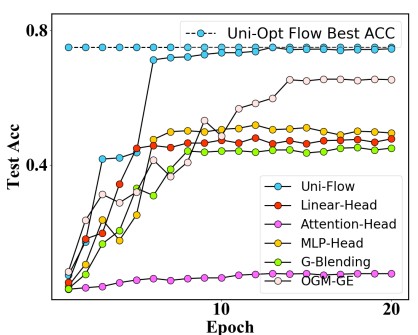

(a) RGB encoder evaluation on VGG-Sound.  (b) Optical flow encoder evaluation on UCF101.

Figure 2: By building a linear classifier on encoders and checking the top-1 accuracy, we evaluate the RGB encoder in VGG-Sound and the optical flow encoder in UCF101 from different multi-modal late-fusion methods.

allows interactions between modalities to learn representations beyond uni-modal features, namely paired features. Based on this, we offer guidance on multi-modal late-fusion learning. Finally, we provide a theoretical analysis of Modality Laziness and justification for our solution.

**Discussion.** The importance of uni-modal prior varies across different multi-modal tasks. In tasks like video classification (Chen et al., 2020b) and action recognition (Feichtenhofer et al., 2016; Wang et al., 2020), uni-modal models can achieve good performance alone, suggesting that uni-modal priors in these settings are essential. Visual question and answering (VQA) (Agrawal et al., 2018) is a counter example. Specifically, the same image with different text questions may have totally different labels, making it pointless to check its uni-modal accuracy. In this paper, *we focus on the tasks where uni-modal priors are essential*, following Wu et al. (2022); Peng et al. (2022).

### 3.1 INSUFFICIENT LEARNING OF UNI-MODAL FEATURES IN MULTI-MODAL TRAINING

This subsection illustrates that existing multi-modal late-fusion training methods suffer from Modality Laziness. Even recent methods, G-Blending (Wang et al., 2020) and OGM-GE (Peng et al., 2022), are no exception.

**In multi-modal late-fusion learning**, each modality is encoded by its corresponding encoder and then a fusion module is applied on top of them to produce outputs. By building a classifier on frozen encoder (Chen et al., 2020c), we assess the learned representations of the encoder:

- As Table 1 shows, all encoders from multi-modal training are worse than those from uni-modal training, especially the RGB encoder in VGG-Sound and optical flow encoder in UCF101. No matter which optimizer is used (Appendix A.3).

Table 2: Top-1 test accuracy of multi-modal models and uni-modal models on certain classes of VGG-Sound. Avg Pred: average the two uni-modal models' predictions directly. Linear Clf: train a multi-modal classifier on top of uni-modal trained encoders. Naive Fusion: train a multi-modal late-fusion model from scratch.

| Class ID | 164 | 303 | 33 | 255 | 91 | 4 | 152 | 127 | 68 | 155 | mean acc |
|---|---|---|---|---|---|---|---|---|---|---|---|
| Uni-RGB | 3 | 2 | 4 | 3 | 4 | 12 | 2 | 0 | 15 | 5 | 5 |
| Uni-Audio | 30 | 7 | 34 | 10 | 43 | 50 | 18 | 0 | 53 | 32 | 27.7 |
| Avg Pred | 37 | 10 | 37 | 7 | 28 | 63 | 21 | 0 | 51 | 30 | 28.4 |
| Linear Clf | 35 | 15 | 33 | **27** | **60** | 65 | 26 | 2 | 53 | **49** | 36.5 |
| Naive Fusion | **43** | **18** | **48** | 22 | 55 | **67** | **26** | **4** | **72** | 40 | **39.5** |

---

**Algorithm 1** Uni-Modal Teacher (UMT) for late-fusion learning

---

**Input:** Uni-modal supervised pre-trained models $F_{pretrain}^{m_1}, F_{pretrain}^{m_2}$, random initialized late-fusion multi-modal model $F^{mm}$, iteration number $N$, loss weight $\lambda_{task}, \lambda_{distill}$.

**for** 0 **to** $N$ **do**

    Sample multi-modal data $\{X^{m_1}, X^{m_2}, Y\} \sim \mathcal{D}$.

    Compute uni-modal pre-trained features $f_{pre}^{m_1}, f_{pre}^{m_2}$ of the data by $F_{pretrain}^{m_1}, F_{pretrain}^{m_2}$.

    Compute the prediction and features $\hat{Y}, f^{m_1}, f^{m_2}$ from multi-modal model.

    Compute the losses between $\hat{Y}, f^{m_1}, f^{m_2}$ and $Y, f_{pre}^{m_1}, f_{pre}^{m_2}$ and multiply by the $\lambda_{task}, \lambda_{distill}, \lambda_{distill}$, respectively.

    Update the multi-modal model by SGD or its variant.

**end for**

**Return:** A multi-modal model trained by UMT.

---

- As Figure 2 shows, throughout the training process, the two encoders mentioned above not only cannot achieve comparable performance to their uni-modal counterparts but are far worse than them.

## 3.2 How does a Model Benefit from Multi-modal Training?

In Sec 3.1, we empirically show that recent late-fusion methods suffer from insufficient learning of uni-modal features. Combining predictions from uni-modal trained models avoids laziness by nature, which raises another question: *How does a multi-modal model benefit from multi-modal training?* We answer this question by investigating different models on VGG-Sound and find that the model learns some representations beyond uni-modal features.

As Table 2 shows, in certain classes of VGG-Sound, the accuracy of naive fusion (navie fusion or naive multi-modal late-fusion learning means no carefully designed tricks are used) exceeds the sum of the accuracy of the two uni-modal models. Besides, we evaluate two other methods. One is to directly average the uni-modal models' predictions, which has little cross-modal interaction. The other one is to train a multi-modal linear classifier on top of uni-modal pre-trained encoders, where modalities can interact with each other through the linear layer. We find that naive fusion training, which owns maximum freedom of cross-modal interaction among these models, gets the best mean accuracy across these classes, suggesting that joint training enables the model to learn representations beyond uni-modal features, which we term as *paired features*. They are a type of feature that uni-modal training cannot learn.

We offer more explanations on paired features in Appendix A.10. We also analyze more datasets and find that different datasets have different characteristics, we put more experimental analysis and interpretation in Appendix A.7.

## 3.3 Guidance on Multi-modal Learning

Given a multi-modal task, if both uni-modal features and paired features are essential, **U**ni-**M**odal **T**eacher (UMT) is effective; if both modalities have strong uni-modal features and paired features

are not important enough, simply combining the predictions of uni-modal models works well, which is named as **Uni-Modal Ensemble** (UME).

**UMT.** **U**ni-**M**odal **T**eacher (UMT) is proposed for late-fusion joint training. It distills the pre-trained uni-modal features to the corresponding parts of multi-modal late-fusion models. Distilling knowledge from uni-modal models can help multi-modal models learn uni-modal features better, which happens in feature-level. The framework of UMT is shown in Algorithm 1 and Figure 4. Noting that we use the same backbone in uni-modal and multi-modal model for a specified modality. The backbones and loss function used can be found in Sec 4. More details can be found in Appendix A.4.

**UME.** If both modalities have strong uni-modal features, joint training does more harm than good. Combining predictions of uni-modal models avoids insufficient learning of uni-modal features by nature. Firstly, we can train uni-modal models independently. Then, we can give final output by weighting the predictions of uni-modal models. The simple ensemble method is named as **Uni-Modal Ensemble** (UME). We demonstrate that UME can show competitive performance on certain multi-modal datasets.

**An empirical trick to decide which method to use.** We can train a multi-modal linear classifier on uni-modal pre-trained encoders and compare that with averaging predictions of uni-modal models. If the performance of the classifier is better, it means we can benefit from cross-modal interaction in this task and we can choose UMT, where cross-modal interactions are preserved while guaranteeing improved learning of uni-modal features; otherwise, the simple cross-modal interaction does more harm than good because of the strong uni-modal features of each modality, and we can choose UME, which avoids Modality Laziness completely.

### 3.4 THEORETICAL CHARACTERIZATION AND JUSTIFICATION

In this subsection, we characterize Modality Laziness of Sec 3.1 from a feature learning perspective and prove it does hurt the generalization of the model. And then, we give justification for the learning guidance proposed in Sec 3.3.

Before diving into the technical details, we first provide some intuition behind the proof. Our goal is to show that how Modality Laziness happens in multi-modal joint training, and we refer to Figure 3 as an illustration. Here we omit the effect of paired features for easier to understand the intuition. During the naive multi-modal training process, learning those easy-to-learn features suffices to reach zero training error (point A in Figure 3). However, the model is under-trained at point A, and the zero-training-error region stops us from further training. As a comparison, uni-modal models can learn more features and achieve point B, outperforming point A.

We next give Modality Laziness a theoretical explanation under a simple but effective regime. We mainly consider cases with two modalities $x^{m_1}$ and $x^{m_2}$, and *similar techniques can be directly generalized to the cases with more modalities*.

**Data distribution.** We formalize the distribution of the multi-modal features. Specifically, we abstract the features into uni-modal features (Definition 3.1) and paired features (Definition 3.2) to describe the core differences between uni-modal training and multi-modal joint training. We consider the binary classification regime where the label $y$ has a uniform distribution over $\{-1, 1\}$ without loss of generality. Such simplification is self-contained to describe the differences between uni-modal features and paired features.

**Definition 3.1** (Uni-modal features, *which can be learned from uni-modal training*)**.** The $i$-th uni-modal feature ($f_i(x^{m_1})$) in modality $x^{m_1}$ is generated as[3]:

$$\text{w.p. } p(f_i), \ yf_i(x^{m_1}) > 0;$$
$$\text{w.p. } 1 - p(f_i) - \epsilon(f_i), \ yf_i(x^{m_1}) = 0;$$
$$\text{w.p. } \epsilon(f_i), \ yf_i(x^{m_1}) < 0.$$

The $i$-th uni-modal feature ($g_i(x^{m_2})$) in modality $x^{m_2}$ is similarly generated with parameters $p(g_i)$ and $\epsilon(g_i)$.

---

[3]We simplify "with probability" as "w.p."

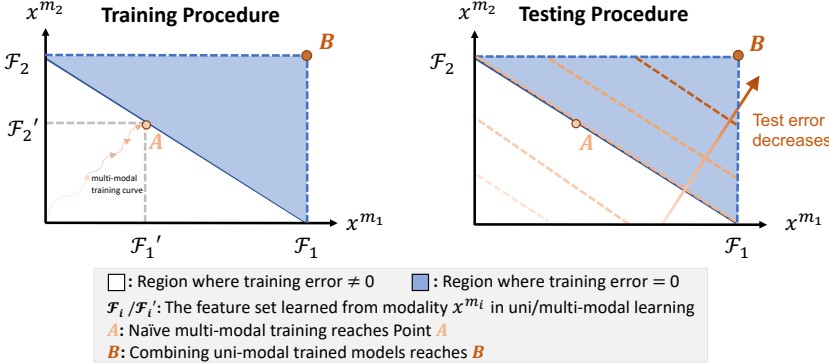

Figure 3: An illustration of the feature learning results of uni-modal training and multi-modal training without considering paired features. In uni-modal training, modality $x^{m_i}$ learns feature set $\mathcal{F}_i$. However, naive joint training learns less features of each modality than uni-modal training when getting zero training error (namely $\mathcal{F}_i'$). Uncontroversially, combining predictions of individual trained uni-modal models (B) outperforms naive joint training (A).

**Definition 3.2** (Paired features, *which can only be learned from cross-modal interaction*). The $j$-th paired feature[4] $h_j$ is generated as:

$$\text{w.p. } p(h_j), \ yh_j(x^{m_1})h_j(x^{m_2}) > 0;$$
$$\text{w.p. } 1 - p(h_j) - \epsilon(h_j), \ yh_j(x^{m_1})h_j(x^{m_2}) = 0;$$
$$\text{w.p. } \epsilon(h_j), \ yh_j(x^{m_1})h_j(x^{m_2}) < 0.$$

When the context is clear, we abuse the notation $r_i$ to represent either $f_i$ (uni-modal feature in modality $x^{m_1}$), $g_i$ (uni-modal feature in modality $x^{m_2}$), or $h_i$ (paired feature). We name $p(r_i)$ as the *predicting probability* of feature $r_i$. When $r_i$ is present (meaning that $r_i \neq 0$), we use $\mathbb{I}(r_i > 0) - \mathbb{I}(r_i < 0)$ to predict $y$. Otherwise ($r_i = 0$), we random guess $y$ uniformly over $\{-1, 1\}$. To simplify the discussion, we always assume $\epsilon(f_i) = p(f_i)/c$, where $c > 1$ is a fixed constant. For the ease of notations, we define the empty feature in Definition 3.3.

**Definition 3.3** (Empty Feature). Empty feature $e_i$ is a kind of uni-modal feature (or paired feature) with $p(e_i) = \epsilon(e_i) = 0$.

**Evaluation procedure.** When the context is clear, we abuse $r_i$ to denote the learned features. For each data point, we random guess $\hat{y}$ on $\{-1, 1\}$ uniformly when $\sum_i \mathbb{I}(r_i > 0) = \sum_i \mathbb{I}(r_i < 0)$. Otherwise, we predict the label by $\hat{y} = 2\mathbb{I}(\sum_i \mathbb{I}(r_i > 0) > \sum_i \mathbb{I}(r_i < 0)) - 1$. We define the error as $\sum_i \mathbb{I}(yr_i < 0) - \sum_i \mathbb{I}(yr_i > 0)$.

**Training procedure.** (a.) *multi-modal joint training*, which directly train the model using both modality $x^{m_1}$ and modality $x^{m_2}$; (b.) *uni-modal ensemble*, which firstly train the features via independent training ($x^{m_1}$ and $x^{m_2}$ separately), and then combine the $x^{m_1}$-learned features and $x^{m_2}$-learned features.

During the training process, we first initialize all the features with empty features $e_i$ to imitate random initialization. The models then learn the features in descending order of predicting probability, meaning that the powerful features (with large predicting probability) are learned first[5]. Our goal is to minimize the training error to zero[6].

We now state our main theorem in Theorem 3.4, demonstrating naive joint training learn fewer uni-modal features compared to uni-modal training, which hurts the model's generalization.

**Theorem 3.4.** *In uni-modal ensemble, assume that the training procedure learns $b_{m1}$ features in modality $x^{m_1}$ and learns $b_{m2}$ features in modality $x^{m_2}$. We order the probability of uni-modal fea-*

---

[4]We abuse the notation $h$ to simplify the notations where $h(x^{m_1})$ and $h(x^{m_2})$ can have different forms.

[5]Recent works have demonstrated that neural networks indeed prefer easy-to-learn features (Shah et al., 2020; Pezeshki et al., 2020).

[6]We always assume that the training error can be minimized to zero.

tures (both $x^{m_1}$ and $x^{m_2}$) in decreasing order of predicting probability $p$, namely, $p_{[1]}, p_{[2]}, \ldots$. In multi-modal training approaches, assume that the training procedure learns $k_{m1}$ uni-modal features in modality $x^{m_1}$, learns $k_{m2}$ uni-modal features in modality $x^{m_2}$, and learns $k_{pa}$ paired features with predicting probability $p(h_1), \ldots, p(h_{k_{pa}})$. We provide three types of laziness:

(a. ) **Quantity Laziness**: $k_{m1} + k_{m2} + k_{pa} \leq \min\{b_{m1}, b_{m2}\}$.

(b. ) **Uni-modal Laziness**: *Each modality in multi-modal training approaches performs worse than uni-modal training.*

(c. ) **Performance Laziness**: *Consider a new testing point, then for every $\delta > 0$, if the following inequality holds:*

$$\sum_{i \in [k_{pa}]} p(h_i) \leq \sum_{i \in [b_{m1}+1, b_{m1}+b_{m2}]} p_{[i]} + \Delta(\delta),$$

*where $\Delta(\delta) = \sqrt{8(k_{pa} + b_{m1} - k_{m1} + b_{m2} - k_{m2})\log(1/\delta)}$, then with probability[7] at least $1 - \delta$, uni-modal ensemble outperform multi-modal training approaches concerning the loss on the testing point with probability.*

In theorem 3.4, we describe three notations of laziness problem: **Quantity Laziness** indicates that the number of features learned in naive multi-modal training is less than uni-modal training. **Uni-modal Laziness** shows encoders from multi-modal training perform worse than from uni-modal training because of Quantity Laziness, which fits the experimental results in sec3.1. **Performance Laziness** compares the performance of multi-modal joint training approaches with Uni-Modal Ensemble, demonstrating that when uni-modal features dominate, combining uni-modal predictions is more effective. We defer the complete proof to Appendix B.1 and generalize that to more modalities (Appendix B.2). We give a concrete example in Appendix B.3 to better illustrate Theorem 3.4.

We next prove that UMT proposed in Sec 3.3 indeed helps uni-modal feature learning and can also learn some easy-to-learn paired features in Theorem 3.5 and Appendix B.3.

**Theorem 3.5.** *Denote the paired features by $h_1, \ldots h_L$ with corresponding predicting probability $p(h_1), \ldots, p(h_L)$. Assume that distillation can boost the training priority by $p^0 > 0$. If there exists paired features whose predicting probability exceeds the boosting probability $p^0$, namely, the set $\mathcal{S}$ is not empty:*

$$\mathcal{S} = \{h_i : p(h_i) > p^0\} \neq \phi.$$

*Then UMT helps uni-modal feature learning and can also learn easy-to-learn paired features.*

## 4 EXPERIMENTS

In Sec 3.4, we justify our method theoretically. In this section, we firstly introduce the experimental setup and then demonstrate that choosing a suitable learning method from UMT and UME can outperforming other complex late-fusion or intermediate-fusion methods in various multi-modal tasks.

### 4.1 EXPERIMENTAL SETUP

**Dataset.** We run experiments on four datasets. *Kinetics-400* (Kay et al., 2017) is a video recognition dataset with 240k videos for training and 19k for validation. We treat the two modalities, RGB and audio, as the inputs. *VGG-Sound* (Chen et al., 2020b) is an audio-visual classification dataset which contains over 200k video clips for 309 different sound classes. *UCF101* (Soomro et al., 2012) is an action recognition dataset with 101 action categories, including 7k videos for training and 3k for testing. *ModelNet40* is a 3D object classification task with 9,483 training samples and 2,468 test samples. Following Wu et al. (2022), we treat the front and rear view as two modalities.

**Training Settings.** In VGG-Sound, UCF101 and ModelNet40, we use ResNet as our backbone, all with 18 layers. As for Kinetics-400, we use 50 or 101 layers' ResNet to encode the inputs. Noting that 3D CNN is used for visual data of VGG-Sound and Kinetics-400. The data preprocessing, hyper-parameters, optimizer can be found in the Appendix A.1 and A.2.

---

[7]The probability is taken over the randomness of the testing point

Table 3: Comparison between averaging uni-modal predictions and multi-modal classifier trained on uni-modal pre-trained encoders.

| Dataset | MM Clf | Avg Preds |
|---|---|---|
| VGG-Sound | **51.0** | 46.1 |
| Kinetics-400 | **76.4** | 74.8 |
| UCF101 | 84.4 | **86.8** |
| ModelNet40 | 91.7 | **91.9** |

Table 4: Evaluation of uni-modal classifiers from the multi-modal linear classifier trained on uni-modal pre-trained encoders in UCF101. This evaluation method borrows from Peng et al. (2022).

| Model | RGB | Opt-flow |
|---|---|---|
| Uni-Clf from MM Clf | 68.2 | 52.9 |
| Uni-Modal Model | **77.1** | **75.0** |

Table 5: Results of different late-fusion methods. * means the result comes from its original paper

| Method | VGG-Sound | Kinetics-400 |
|---|---|---|
| Linear-Head | 49.5 | 74.3 |
| MLP-Head | 44.8 | 74.8 |
| Atten-Head | 49.8 | 74.1 |
| Aux-CELoss | 49.9 | 73.2 |
| G-Blending | 50.4 | 75.8* |
| OGM-GE | 50.6* | 74.5 |
| UMT (ours) | **53.5** | **76.8** |

Table 6: Comparison between UMT and Audio-Visual SlowFast (Xiao et al., 2020) on Kinetics-400. AVSlowFast is an representative intermediate fusion method.

| Method | RGB Encoder | Acc |
|---|---|---|
| AVSlowFast | SlowFast-50 | 77.0 |
| UMT (ours) | SlowFast-50 | **78.1** |
| AVSlowFast | SlowFast-101 | 78.8 |
| UMT (ours) | SlowFast-101 | **79.4** |

## 4.2 AN EMPIRICAL TRICK TO DECIDE WHICH LEARNING METHOD TO USE.

We train a multi-modal linear classifier on frozen uni-modal pre-trained encoders and compare that with averaging uni-modal predictions. As Table 3 shows, in VGG-Sound and Kinetics-400, the classifier is better, meaning cross-modal interaction can benefit the classifier in the two datasets. However, in UCF101 and ModelNet40, averaging uni-modal predictions performs well. To explore why the classifier fails in UCF101, we check the uni-modal classifiers of the newly trained multi-modal linear layer (details in Appendix A.7.1). As Table 4 shows, they are far worse than the uni-modal models. The result shows the *simple linear classifier suffers from serious Modality Laziness in UCF101*, which negatively impact the performance. Both modalities in ModelNet40 also have strong uni-modal features and can achieve 89% accuracy individually. Averaging uni-modal predictions avoids laziness problem and achieves competitive performance. Based on the above analysis, we perform UMT on VGG-Sound and Kinetics-400, and UME on UCF101 and ModelNet40.

## 4.3 UMT IS AN EFFECTIVE REGULARIZER FOR UNI-MODAL FEATURE LEARNING

In this subsection, we demonstrate that Uni-Modal Teacher outperforms other multi-modal training methods in VGG-Sound and Kinetics-400. We use MSELoss as the distillation loss and set the weight of that as 50. Cross Entropy is used as classification loss and its weight is set as 1.

### 4.3.1 UMT IS EFFECTIVE ON VGG-SOUND AND KINETICS-400.

**UMT vs Other Late-Fusion Methods.** The late-fusion architecture is commonly used for multi-modal classification tasks (Wang et al., 2020; Peng et al., 2022). In late-fusion architecture, the features are extracted from different modalities by the corresponding encoders, and then the head layer is applied to output predictions. We compare different heads, including linear layer, MLP, and attention layer. In UMT, we use a simple linear layer as the multi-modal head. We also conduct another experiment, which adds extra uni-modal linear heads to receive the uni-modal features and generating additional losses to joint optimize the model, namely Auxiliary-CEloss. Auxiliary-CEloss gives all losses equal weights, while G-Blending reweights the losses according to the overfitting-to-generalization-ratio (OGR) (Wang et al., 2020). OGM-GE (Peng et al., 2022) controls the optimization of each modality by online gradient modulation. However, both OGM-GE and G-Blending are complex to implement. As shown in Table 5, UMT outperforms other methods.

Table 7: Evaluation on the encoders trained by naive multi-modal training and UMT.

| Methods. | VGG-Sound | | Kinetics-400 | |
|---|---|---|---|---|
| | RGB | Audio | RGB | Audio |
| Uni-Train | 23.2 | 45.2 | 74.1 | **23.5** |
| MM Baseline | 15.9 | 18.3 | 72.9 | 18.3 |
| UMT | **24.4** | **45.9** | **74.6** | 21.6 |

Table 8: Self-Distillation vs UMT on VGG-Sound.

| Method | Test Acc |
|---|---|
| Baseline | 49.5 |
| Self-Distill (label) | 49.7 |
| Self-Distill (feature) | 49.9 |
| UMT | **53.5** |

Table 9: Comparison Uni-Modal Ensemble with other joint training methods on UCF101.

| Method | Test Acc |
|---|---|
| Linear-Head | 82.3 |
| MLP-Head | 80.0 |
| Atten-Head | 74.2 |
| Aux-CELoss | 81.3 |
| G-Blending | 83.0 |
| OGM-GE | 84.0 |
| UME (ours) | **86.8** |

Table 10: Comparison between Uni-Modal Ensemble and balanced multi-modal learning algorithm (Wu et al., 2022) on ModelNet40. * means the result comes from Wu et al. (2022).

| Method | Test Acc |
|---|---|
| multi-modal (vanilla) | $90.09 \pm 0.58$* |
| +RUBi | $90.45 \pm 0.58$* |
| +random | $91.36 \pm 0.10$* |
| +guided | $91.37 \pm 0.28$* |
| UME (ours) | $\mathbf{91.92 \pm 0.14}$ |

**UMT vs AVSlowFast.** Audio-Visual SlowFast is an representative intermediate fusion method. We compare UMT with AVSlowFast in Kinetics-400. As Table 6 shows, under different RGB encoders, UMT consistently exceeds AVSlowFast, although we cannot reproduce their results due to the dynamics of Kinetics-400 (Appendix A.11).

**Ablation Study of UMT.** We first evaluate the encoders of UMT by training linear classifiers on them to verify that UMT does improve the uni-modal feature learning. As Table 7 shows, UMT makes its encoders stand out. Benefiting from uni-modal distillation, some encoders even outperform their uni-modal counterparts. We then compare UMT with classic self-distillation methods (distillation on soft label (Hinton et al., 2015) and feature (Romero et al., 2014)). As Table 8 shows, naive self-distillation can only bring limited improvement, showing that UMT improves overall performance by improving the uni-modal feature learning instead of knowledge distillation.

### 4.3.2 UNI-MODAL ENSEMBLE IN MULTI-MODAL LEARNING

In this subsection, we demonstrate that Uni-Modal Ensemble is effective on multi-modal datasets where modalities have strong uni-modal features, outperforming other complex methods. *Even though we don't combine these uni-modal predictions in any special way, but simply average.*

**In UCF101**, we compare Uni-Modal Ensemble with various multi-modal late-fusion methods. As Table 9 shows, although Gradient Blending (Wang et al., 2020) and OGM-GE (Peng et al., 2022) outperforms baseline methods, they are far worse than Uni-Modal Ensemble.

**In ModelNet40**, the main comparing methods come from Wu et al. (2022), which uses a multi-modal DNN with intermediate fusion. It proposes a balanced multi-modal algorithm which balances conditional utilization of each modality by re-balancing the optimization step. UME surpass their balanced multi-modal algorithm, as Table 10 shows.

## 5 CONCLUSION

This paper analyzes the phenomenon of insufficient uni-modal feature learning in multi-modal training and proves that it does hurt the overall performance. We propose to choose proper learning method from Uni-Modal Ensemble and proposed Uni-Modal Teacher according to the distribution of uni-modal and paired features and demonstrate the effectiveness of the guiding principle.

## 6 REPRODUCIBILITY STATEMENT

In our paper, we detail the specific implementation steps of the methods in Sec 3.3 and also give the values of the hyper-parameters we use in detail in Appendix A.2. We also provide the codes in supplementary material, which can reproduce the results of the experiments.

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

## A    EXPERIMENTAL DETAILS AND ADDITIONAL EXPERIMENTS

### A.1    DATASETS

Here, we describe the preprocessing of Kinetics-400, VGG-Sound, UCF101 and ModelNet40 in detail.

**Kinetics-400** dataset (Kay et al., 2017) contains over 240k videos for training and 19k for validation, which we download from cvdfoundation [8]. Kinetics-400 is a commonly used dataset with 400 classes, and we mainly follow the open source preprocessing methods to process that. For RGB modality, we follow the procedure of PySlowFast [9], which resizes the video to the short edge size of 256. and for audio modality, we follow mmaction2 [10] to extract specgram features. When performing joint training, we take consecutive 64 frames from a video with fps of 30 and random crop the video to 224*224, and for audio inputs, we take the specgram that can be aligned in time with the clip extracted from the video. When testing, we ensemble the predictions from uniformly sampled clips with RGB and audio from a video and give the final outputs, following PySlowfast.

**VGG-Sound** dataset (Chen et al., 2020b), which contains over 200k video clips for 309 different sound classes, is also used for evaluating our method. It is an audio-visual dataset *in the wild* where each object that emits sound is also visible in the corresponding video clip, making it suitable for scene classification tasks. Please note that some clips in the dataset are no longer available on YouTube, and we actually use about 175k videos for training and 15k for testing, but the number of classes remains the same. We design a preprocessing paradigm to improve training efficiency as follows: (1) each video is interpolated to $256\times256$ and saved as stacked images; (2) each audio is first converted to 16 kHz and 32-bit precision in the floating-point PCM format, then randomly cropped or tiled to a fixed duration of 10s. For video input, 32 frames are uniformly sampled from each clip before feeding to the video encoder. While for the audio input, a 1024-point discrete Fourier transform is performed using nnAudio (Cheuk et al., 2020), with 64 ms frame length and 32 ms frame-shift. And we only feed the magnitude spectrogram to the audio encoder.

**UCF101** dataset (Soomro et al., 2012) is an action recognition dataset with 101 action categories, including 7k videos for training and 3k for testing. And we use the rgb and flow provided by (Feichtenhofer et al., 2016). For RGB, we use one image of $(3*224*224)$ as the input; while for flow, we use a stack of optical flow images which contained 10 x-channel and 10 y-channel images, So its input shape is $(20*224*224)$. During training, we perform random crop and random horizontal flip as the data augmentation; while testing, we resize the image to 224 and do not perform data augmentation operations.

**ModalNet40** is a 3D object classification dataset with 9,483 training samples and 2,468 test samples. We base on the front view and the rear view of the 3D object to classify that, following Wu et al. (2022).

### A.2    TRAINING HYPERPARAMETERS

In this subsection, we show the hyperparameters of our experiments in UCF101 and VGG-Sound in Table 11.

As for Kinetics-400's RGB modality, we totally follow the hyperparameters and settings of PySlow-Fast [11]. As for audio modality, we modify the hyperparameters [12] to be as consistent as possible with the RGB training for further joint training. Specifically, we use the same learning rate and batch size as RGB training used.

As for ModelNet40, we totally follow the experimental settings of Wu et al. (2022) [13].

---

[8] https://github.com/cvdfoundation/kinetics-dataset

[9] https://github.com/facebookresearch/SlowFast/

[10] https://github.com/open-mmlab/mmaction2/blob/master/tools/data/build_audio_features.py/

[11] https://github.com/facebookresearch/SlowFast/configs/Kinetics/SLOWFAST_8x8_R50.yaml

[12] openmmlab/mmaction2/blob/master/configs/recognition_audio/resnet/tsn_r18_64x1x1_100e_kinetics400_audio_feature.py

[13] https://github.com/nyukat/greedy_multimodal_learning

Table 11: The Hyperparameters used in our experiments for VGG-Sound and UCF101.

| Hyperparameter | Value (VGG-Sound) | Value (UCF101) |
|---|---|---|
| Encoder | ResNet3D (Video), 2D (Audio) | ResNet2D(Both Modalities) |
| Linear Head | (1024, 309) | (1024, 101) |
| MLP Head | (1024, 1024) | (1024, 1024) |
| | ReLU | ReLU |
| | (1024, 309) | (1024, 101) |
| Attension Head | Attension Layer (without new parameters) + a linear layer | |
| Training Epoches | 20 | 20 |
| LR | 1e-3 | 1e-2 |
| Batch Size | 24 | 64 |
| Optimizer | Adam | SGD |
| Scheduler | StepLR (step=10, gamma=0.1) | ReduceLROnPlateau (patience=1) |
| Loss Fusion | Cross Entropy for task, MSE for distillation | |

## A.3  CAN EXISTING OPTIMIZERS SOLVE MODALITY LAZINESS?

Table 12: Top-1 test accuracy (in %) of linear classifiers trained on frozen encoders from multi-modal late-fusion training under different optimizers and uni-modal training on VGG-Sound.

| Optimizer | Multi-modal Performance | Audio Encoder | RGB Encoder |
|---|---|---|---|
| SGD | 47.13 | 40.02 | 15.53 |
| RMSprop | 47.90 | 42.77 | 13.64 |
| Adagrad | 42.19 | 35.68 | 19.65 |
| Adadelta | 23.18 | 17.70 | 17.37 |
| Adamw | 49.39 | 42.41 | 15.11 |
| Adam | 49.47 | 43.44 | 15.56 |
| Uni-Training | / | **45.15** | **23.17** |

While the results in Table 1 show that different multi-modal methods suffer from learning insufficient uni-modal features, *how about changing the optimizer*? To answer this question, we try different optimizers for multi-modal late-fusion training (with a linear multi-modal head), including SGD, RMSprop, Adagrad, Adadelta, Adamw and Adam. As Table 12 shows, Modality Laziness exists no matter which optimizer is used.

## A.4  DETAILS ON UNI-MODAL TEACHER (UMT)

In this subsection, we describe how Uni-Molda Teacher (UMT) applies on multi-modal late-fusion tasks. The overall architecture can be found in Figure 4.

**UMT in late-fusion classification.** In multi-modal late-fusion architecture, modalities are first encoded by the corresponding encoders and then mapped to the output space by a multi-modal fusion head (Figure 4 left). Uni-Modal Teacher distills the pre-trained uni-modal features to the corresponding parts in multi-modal networks in multi-modal training (Figure 4 right). Uni-modal distillation happens before fusion, so it's suitable for late-fusion multi-modal architecture. The pre-trained uni-modal features are generated by inputting the data to the pre-trained uni-modal models.

**UMT's weights.** For VGG-Sound and Kinetics, we use 50 (both audio feature distillation and RGB feature distillation) as the distillation loss's weight. We test different distillation weights on VGG-Sound and Kinetics-400. As shown in Table 13, on both datasets, UMT performs well with an distillation weight of 50.

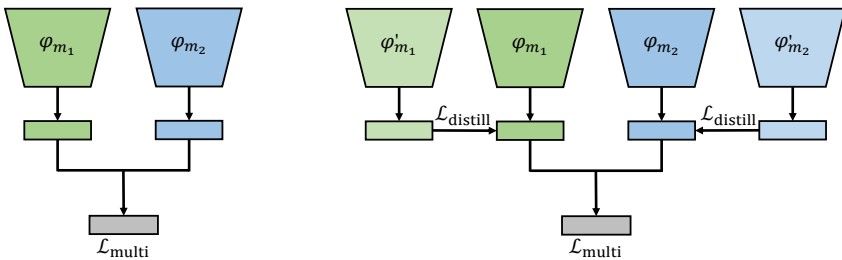

Figure 4: Model architecture of naive late fusion (left) and Uni-Modal Teacher (UMT) (right). $\varphi'_{m_i}$ is the encoder which is supervised pre-trained on uni-modal data. $\varphi_{m_i}$ is a random initialed encoder without pre-training. $\mathcal{L}_{multi}$ is the loss between multi-modal predictions and labels. $\mathcal{L}_{distill}$ is the uni-modal distillation loss.

Table 13: Different distillation weights of UMT on VGG-Sound and Kinetics-400

| Dataset | 0 | 1 | 10 | 20 | 50 | 100 |
|---|---|---|---|---|---|---|
| VGG-Sound | 49.46 | 49.51 | 51.31 | 51.51 | **53.46** | 53.11 |
| Kinetics-400 | 74.25 | 74.99 | 75.57 | 76.11 | **76.77** | 76.55 |

## A.5 DROPOUT IN MULTI-MODAL TRAINING.

Here we consider the common regularizer, dropout (Srivastava et al., 2014), and a variant of it, namely modality-wise dropout, which randomly drops (with probability $1/3$) the feature from one modality in each iteration. Modality dropout is akin to the ModDrop in Neverova et al. (2015). As Table 14 shows, modality-wise dropout is significantly better than dropout, which implies that modality-wise laziness is serious and modality-wise dropout is also effective.

## A.6 FINETUNING THE UNI-MODAL PRE-TRAINED ENCODERS

In this subsection, we use the uni-modal pre-trained encoders' parameters as the initialized weights in multi-modal training and randomly initialize a multi-modal linear classifier on the encoders. We set the classifier's learning rate as $1e-3$ and try different learning rates on the encoders.

As Table 15 shows, using the uni-modal supervised pre-trained encoder's weights in multi-modal training and then fine-tuning the whole multi-modal model can bring some improvement compared to naive fusion (49.46) but is worse than UMT, which gets 53.46 accuracy. When the learning rate of encoders is large, the encoders forget some abilities to extract uni-modal features.

Ngiam et al. (2011) proposes to use Bimodal Deep Autoencoder to pre-train the encoders with multiple modalities. It is a direction worth exploring to address Modality Laziness of deep multi-modal models.

## A.7 THE ROLE OF CROSS-MODAL INTERACTION ON DIFFERENT DATASETS

In this subsection, we conduct various experiments to further investigate the effect of cross-modal interaction and explore the benefits and harms that cross-modal interaction brings in different multi-

Table 14: Dropout in multi-modal training on VGG-Sound.

| Method | Performance |
|---|---|
| Baseline | 49.46 |
| Dropout | 49.83 |
| Modal-Drop | 51.37 |
| UMT | **53.46** |

Table 15: The top-1 test accuracy of finetuning the uni-modal pre-trained encoders and linear evaluation on finetuned encoders on VGG-Sound.

| Encoder LR | Top-1 Acc | Encoder Eval | |
| | | Audio | RGB |
| --- | --- | --- | --- |
| 1e-3 | 50.98 | 43.98 | 21.86 |
| 1e-4 | 49.37 | 44.71 | 21.97 |
| 1e-5 | 50.45 | 45.28 | 23.13 |
| 1e-6 | 50.86 | 45.29 | 23.27 |
| 0 | 50.95 | 45.15 | 23.17 |

modal tasks/datasets. We find that in different datasets, cross-modal interaction has a different effect on the performance.

### A.7.1 AVERAGING THE UNI-MODAL PREDICTIONS *vs* THE LINEAR CLASSIFIER TRAINED ON UNI-MODAL PRE-TRAINED ENCODERS.

In Sec 4.2, we train a multi-modal linear classifier on frozen uni-modal pre-trained encoders and compare this classifier with directly averaging uni-modal models' predictions. As Table 3 shows, this classifier does not consistently outperform simply averaging the uni-modal predictions on all datasets. It shows better performance on VGG-Sound and Kinetics-400, but worse performance on UCF101 and ModelNet40.

To further explain this phenomenon, we check and disassemble this new trained multi-modal classifier on UCF101. In the late-fusion multi-modal training, the features of different modalities are concatenated first and then the multi-modal classifier receives them and output predictions. Different modalities in the classifier do not share the parameters. So we split the new trained multi-modal linear classifier into uni-modal classifiers. We use the uni-modal pre-trained encoders to extract features and then the uni-modal classifiers receive the corresponding features and output predictions. Noting that OGM-GE (Peng et al., 2022) uses similar technique to check how well different modalities are trained. As Table 4 shows, the uni-modal classifiers from new trained multi-modal classifiers are significantly worse than uni-modal models, implying that the multi-modal classifier trained on uni-modal pre-trained encoders suffers from serious Modality Laziness on UCF101, although it is just a simple linear layer, resulting in worse performance than directly averaging the uni-modal predictions.

### A.7.2 CLASS-LEVEL EVALUATION ON DIFFERENT MULTI-MODAL DATASETS

In this subsection, we compare naive late-fusion learning with averaging predictions of uni-modal models in *class* level. It's obvious that there are more cross-modal interactions in naive fusion.

Although naive fusion suffers from learning insufficient uni-modal features, we find in some classes in Kinetics and VGG-Sound, the accuracy of naive fusion model outperforms averaging the uni-modal models' predictions, and even outperforms the sum of the accuracy of the two uni-modal models in VGG-Sound and Kinetics-400, as shown in Table 2 and 16.

However, We cannot find any class that naive fusion can exceed the sum of the accuracy of the uni-RGB model and uni-flow model in UCF101. We select classes in UCF101 by sorting the differences of accuracy between naive fusion and the best uni-modal model in class level and the top ten with the largest difference are selected. In these classes where naive fusion has advantages, averaging the predictions can outperform naive fusion in some classes (ID:29, 67, 71), and this phenomenon is not found in VGG-Sound and Kinetics. And as Tabla 4 and Table 17 show, both RGB and optical flow in UCF101 can get strong performance individually. All the evidence shows that in UCF101, the uni-modal features are totally dominate and any joint training can lead to serious Modality Laziness.

**The mapping betwenn class ID and class name in different datasets** The correspondence between id and name of the selected class in VGG-Sound is: 164: People Sniggering, 303: Wood Thrush Calling, 33: Cat Meowing, 255: Sea Waves, 91: Footsteps On Snow, 4: Alligators

Table 16: Top-1 test accuracy of different models on some classes of Kinetics. The accuracy of naive fusion model outperforms averaging the uni-modal models' predictions, and even outperforms the sum of the accuracy of the uni-audio model and uni-video model.

| Class ID | 53 | 90 | 184 | 2 | 368 | 158 | 113 | 263 | 287 | 4 | mean accuracy |
|----------|----|----|-----|---|-----|-----|-----|-----|-----|---|---------------|
| Uni-Audio | 0 | 0 | 0 | 4 | 0 | 0 | 0 | 0 | 4 | 2 | 1 |
| Uni-RGB | 42 | 50 | 22 | 28 | 39 | 43 | 29 | 82 | 76 | 50 | 46.1 |
| Avg Pred | 42 | 50 | 22 | 28 | 39 | 43 | 29 | 82 | 78 | 50 | 46.3 |
| Naive Fusion | **56** | **62** | **32** | **40** | **45** | **49** | **35** | **84** | **86** | **58** | **54.7** |

Table 17: Top-1 test accuracy of different models on selected classes of UCF101. We select the top-10 classes according to the gap of accuracy between the multi-modal and uni-modal models. As we can see, uni-modal model's performance is high, meaning paired features in UCF101 are rare.

| Class ID | 6 | 10 | 12 | 22 | 29 | 31 | 48 | 57 | 67 | 71 | mean accuracy |
|----------|---|----|----|----|----|----|----|----|----|----|---------------|
| Uni-RGB | 74 | 84 | 76 | 61 | 67 | 32 | 78 | 21 | 75 | 57 | 62.5 |
| Uni-Flow | 60 | 82 | 58 | 47 | 61 | 41 | 64 | 24 | 78 | 63 | 57.8 |
| Avg Pred | 70 | 95 | 79 | 64 | **86** | 46 | 89 | 36 | **98** | **83** | 74.6 |
| Naive Fusion | **86** | **95** | **87** | **72** | 83 | **59** | **92** | **42** | 88 | 73 | **77.7** |

Crocodiles Hissing, 152: People Gargling, 127: Mynah Bird Singing, 68: Door Slamming, 155: People Humming.

For Kinetics-400, we sort the classes alphabetically from smallest to largest according to the class name, and then we can get the mapping between class name and id.

In UCF101, the mapping can be found in classInd.txt, a given file of UCF101.

## A.8    CROSS-MODAL INTERACTION IN UNI-MODAL TEACHER (UMT)

In order to verify whether the multimodal loss in UMT makes sense, we train uni-modal models by knowledge distillation to get better performance than encoders trained by UMT and then combine them by introducing a new multi-modal classifier on these encoders. As Table 18 shows, UMT works better in multi-modal performance, although the encoders of UMT in uni-modal evaluation are worse, showing that UMT indeed benefits from cross-modal interaction.

Table 18: Comparison of UMT with combining uni-modal models trained by distillation on VGG-Sound.

| Method | RGB | Audio | R+A |
|--------|-----|-------|-----|
| Linear Clf | **25.99** | **46.00** | 52.98 |
| UMT | 24.43 | 45.89 | **53.46** |

## A.9    EXPLORING
## UMT FOR MULTI-MODAL SEGMENTATION

**NYU Depth V2** dataset (Silberman et al., 2012) contains 1449 indoor RGB-Depth data totally and we use 40-class label setting. The number of training set and testing set is 795 and 654 respectively. All perprocessing operations are following (Seichter et al., 2020).

In contrast to the late fusion classification task, the RGB-Depth semantic segmentation belongs to middle fusion. The main encoder receives RGB inputs, and the depth inputs are fed into the depth encoder. At each intermediate layer, the main encoder fuses its own intermediate outputs and the depth features obtained from the depth encoder, which makes it a mid-fusion task (Seichter et al., 2020). Since features generated by each layer matter, we distill multi-scale depth feature maps using the MSE loss. For feature maps from the RGB encoder, however, since they are generated by fusing RGB and depth modalities, we cannot distill RGB feature maps directly like depth feature maps. To mitigate this effect, we curate predictors, namely 2 layers CNNs, aiming to facilitate the fused feature maps to predict the RGB feature maps trained by the RGB modality before distillation. The

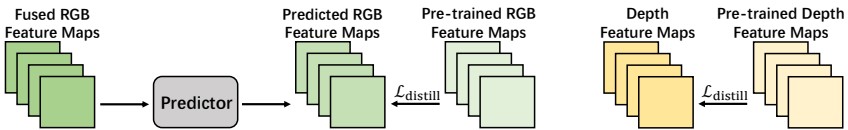

Figure 5: Distillation details of UMT for RGB (left) and depth (right) modalities in multi-modal semantic segmentation (based on ESANet).

Table 19: Model performance comparison under UMT and ESANet on NYU-DepthV2 RGB-Depth semantic segmentation task.

| Initialization | Training Setting | |
|---|---|---|
| | ESANet | UMT |
| From Scratch | 38.59 | **40.45 (+1.86)** |
| ImageNet Pre-train | 48.48 | **49.39 (+0.91)** |

full schematic diagram is presented in Figure 5. As shown in Table 19, UMT can also improve multi-modal segmentation whether the encoder is pre-trained on ImageNet or not.

### A.10 EXPLANATIONS ON PAIRED FEATURES

We revisit the definitions of uni-modal features and paired features: **uni-modal features**, which can be learned by uni-modal training; **paired features**, which can only be learned by cross-modal interaction in joint training. Different datasets contain different proportions of these features.

In this subsection, we use synthetic datasets to explain the uni-modal features and paired features in multi-modal tasks.

**Understanding different types of features in multi-modal tasks by synthetic datasets.** Three different multi-modal datasets are generated to help us understand the uni-modal features and paired features in multi-modal tasks. The process of data generation mainly refers to Hessel & Lee (2020).

**Firstly, we generate a dataset where each modality can extract the features to give correct predictions.** We name this dataset as **Dataset** $\alpha$. The data generation process is as follows:

1. Sample random projection $P_1 \in \mathbb{R}^{d_1 \times d}$ and $P_2 \in \mathbb{R}^{d_2 \times d}$ from $U(-0.5, 0.5)$.
2. Sample $z \in \mathbb{R}^d \sim \mathcal{N}(0, 1)$. Normalize $z$ to unit length
3. Sample $x \in \mathbb{R}^d \sim \mathcal{N}(0, 1)$. Normalize $x$ to unit length
4. if $|x \cdot z| \leq 0.1$, return to the Step 3.
5. If $x \cdot z > 0.1$, then $y = 1$; else $y = 0$.
6. Get the data point $(P_1 x, P_2 x, y)$.
7. If the amount of data generated is less than $N$, return to the Step 3; else break

The $P_1 x, P_2 x$ represents two modalities of the multi-modal dataset and we set $d_1, d_2, d, N$ as $200, 100, 50, 5000$, respectively. And we randomly split 80% of the generated data as train set, and the rest serves as a test set. In this dataset, uni-modal models can extract useful features to give correct predictions and multi-modal joint training is not necessary, as Table 20 shows (**Dataset** $\alpha$). *We name the features, that uni-modal models can learns to give correct predictions in the given task, as uni-modal features.*

**Secondly, we generate another dataset where the model must rely on both the two modalities to make correct predictions.** We name this dataset as **Dataset** $\beta$. The data generation process is as follows:

1. Sample random projection $P_1 \in \mathbb{R}^{d_1 \times d}$ and $P_2 \in \mathbb{R}^{d_2 \times d}$ from $U(-0.5, 0.5)$.

Table 20: Test Accuracy of uni-modal models and multi-modal model on different synthetic datasets. **Synthetic Dataset** $\alpha$ mainly contains uni-modal features which can be learned in uni-modal training; **Synthetic Dataset** $\beta$ mainly contains paired features which need joint training to learn; **Synthetic Dataset** $\gamma$ contains uni-modal features and paired features.

| Dataset | Synthetic Dataset $\alpha$ | Synthetic Datset $\beta$ | Synthetic Dataset $\gamma$ |
|---|---|---|---|
| Uni-modal (Modality 1) | 100 | 51.4 | 70.9 |
| Uni-modal (Modality 2) | 100 | 51.8 | 70.1 |
| Multi-modal | 100 | 92 | 94.4 |

Table 21: The confusion matrix of uni-modal model in **Dataset** $\gamma$. In the data labeled 0, each modality contains features that can give a correct prediction, while in the data labeled 1 or 2, we need both modalities together to make the right predictions.

|  | | Predicted | | |
|---|---|---|---|---|
| | | 0 | 1 | 2 |
| Actual | 0 | 100% | 0 | 0 |
| | 1 | 0 | 57% | 43% |
| | 2 | 0 | 45.4% | 54.6% |

2. Sample $x_1, x_2 \in \mathbb{R}^d \sim \mathcal{N}(0, 1)$. Normalize $x_1, x_2$ to unit length

3. if $|x_1 \cdot x_2| \leq 0.25$, return to the Step 2.

4. If $x_1 \cdot x_2 > 0.25$, then $y = 1$; else $y = 0$.

5. Get the data point $(P_1 x_1, P_2 x_2, y)$.

6. If the amount of data generated is less than $N$, return to the Step 2; else break

This multi-modal dataset is different from the first dataset, because the labels in this dataset are highly dependent on the relationship between the two modalities. As we can see in Table 20 (**Dataset** $\beta$), the uni-modal models can only gives about 50 percent accuracy, while the multi-modal models can give about 90 percent accuracy. In binary classification tasks, 50 percent accuracy is no different from guessing. *In this dataset, because labels are heavily relied on the relationships of the two modalities, we must train both modalities simultaneously to extract the joint representations to learn the relationship of the two modalities, which are beyond uni-modal features. In order to better carry out theoretical analysis, we abstract these representations into paired features, which can only be learned from multi-modal joint training in multi-modal tasks.*

**Finally, we generate a dataset that contains both uni-modal features and paired features.** We name this dataset as **Dataset** $\gamma$. The data generation process is as follows:

1. Sample random projection $P_1 \in \mathbb{R}^{d_1 \times d}$ and $P_2 \in \mathbb{R}^{d_2 \times d}$ from $U(-0.5, 0.5)$.

2. Sample $z \in \mathbb{R}^d \sim \mathcal{N}(0, 1)$. Normalize $z$ to unit length

3. Sample $x \in \mathbb{R}^d \sim \mathcal{N}(0, 1)$. Normalize $x$ to unit length

4. if $x \cdot z \geq 0.1$, get the data point $(P_1 x, P_2 x, y = 0)$. else, return to the step 3 until collecting 2500 data points.

5. Sample $x_1, x_2 \in \mathbb{R}^d \sim \mathcal{N}(0, 1)$. Normalize $x_1, x_2$ to unit length. If $x_1 \cdot z > -0.1$ or $x_2 \cdot z > -0.1$, resample.

6. if $|x_1 \cdot x_2| \leq 0.25$, return to the Step 5.

7. If $x_1 \cdot x_2 > 0.25$, then $y = 2$; else $y = 1$.

8. Get the data point $(P_1 x_1, P_2 x_2, y)$.

9. If the total amount of data generated is less than 7500, return to the Step 5; else break

In the data labeled 0, each modality contains features that can give a correct prediction, while in the data labeled 1 or 2, we need both modalities together to make the right predictions. To further

understand how uni-modal models give predictions in dataset that containing both uni-modal and paired features, we give the confusion matrix of the uni-modal model. As Table 21 shows, the uni-modal model can give correct predictions for data labeled 0, while for data labeled 1 or 2, it fails and gives a random predictions. Because for data labeled 1 or 2, we need to learn the relationship of the two modalities to give the correct predictions.

In this subsection, we mainly discuss the synthetic multi-modal datasets and in Appendix A.7, we conduct various experiments on real-world multi-modal datasets to help us understand the uni-modal features, paired features and cross-modal interaction in multi-modal training better.

**Training settings on synthetics datasets.** We use a two layer MLP with ReLU as activation function. As for hidden layer, we use 200 dimensions for multi-modal training and 100 dimensions for uni-modal training. We use SGD as the optimizer and the learning rate is 0.2. In each iteration, we use the whole training set to compute the gradients. And we provide the code in supplement materials.

### A.11    UNI-MODAL PERFORMANCE IN KINETICS-400

Kinetics-400 is a dynamic dataset, because videos may be removed from YouTube. In this subsection, we report the uni-modal performance of ours and Xiao et al. (2020)'s on Kinetics-400. As Table 22 shows, we cannot reproduce their uni-modal performance and ours are lower than theirs. But we demonstrate that UMT outperforms AVSlowFast in Sec 4.3.1, which shows UMT's effectiveness.

Table 22: Uni-Modal Performance of ours and Xiao et al. (2020)'s on Kinetics-400

|  | ours | Xiao et al. (2020) |
|---|---|---|
| Uni-Audio | 23.5 | **24.8** |
| Uni-RGB (SlowFast-50) | 74.9 | **75.6** |
| Uni-RGB (SlowFast-101) | 77.2 | **77.9** |

# B  PROOF

## B.1  PROOF OF THEOREM 3.4

**Theorem 3.4.** *In uni-modal ensemble, assume that the training procedure learns $b_{m1}$ features in modality $x^{m_1}$ and learns $b_{m2}$ features in modality $x^{m_2}$. We order the probability of uni-modal features (both $x^{m_1}$ and $x^{m_2}$) in decreasing order of predicting probability p, namely, $p_{[1]}, p_{[2]}, \ldots$. In multi-modal training approaches, assume that the training procedure learns $k_{m1}$ uni-modal features in modality $x^{m_1}$, learns $k_{m2}$ uni-modal features in modality $x^{m_2}$, and learns $k_{pa}$ paired features with predicting probability $p(h_1), \ldots, p(h_{k_{pa}})$. We provide three types of laziness:*

(a. ) **Quantity Laziness**: $k_{m1} + k_{m2} + k_{pa} \leq \min\{b_{m1}, b_{m2}\}$.

(b. ) **Uni-modal Laziness**: *Each modality in multi-modal training approaches performs worse than uni-modal training.*

(c. ) **Performance Laziness**: *Consider a new testing point, then for every $\delta > 0$, if the following inequality holds:*

$$\sum_{i \in [k_{pa}]} p(h_i) \leq \sum_{i \in [b_{m1}+1, b_{m1}+b_{m2}]} p_{[i]} + \Delta(\delta),$$

*where $\Delta(\delta) = \sqrt{8(k_{pa} + b_{m1} - k_{m1} + b_{m2} - k_{m2})\log(1/\delta)}$, then with probability[14] at least $1 - \delta$, uni-modal ensemble outperform multi-modal training approaches concerning the loss on the testing point with probability.*

We prove the theorem, which shows that naive joint training indeed suffers from overfitting issues, meaning that it learns less features compared to uni-modal ensemble.

*Proof.* We first introduce some additional notations used in the proof. We define the features trained in $x^{m_1}$-uni-modal training as $f_1(x^{m_1}), \ldots, f_{b_{m1}}(x^{m_1})$, define the features trained in $x^{m_1}$-uni-modal training as $g_1(x^{m_2}), \ldots, g_{b_{m2}}(x^{m_2})$. Therefore, there are in total $b_{m1} + b_{m2}$ features learned in uni-modal ensemble, namely, $f_1(x^{m_1}), \ldots, f_{b_{m1}}(x^{m_1}), g_1(x^{m_2}), \ldots, g_{b_{m2}}(x^{m_2})$. Besides, We define the features trained in multi-modal training approaches as $f_1(x^{m_1}), \ldots, f_{k_{m1}}(x^{m_1})$, $g_1(x^{m_2}), \ldots, g_{k_{m2}}(x^{m_2}), h_1(x^{m_1}, x^{m_2}), \ldots, h_{k_{pa}}(x^{m_1}, x^{m_2})$. When the context is clear, we omit the dependency of $x^{m_1}, x^{m_2}$ and denote them as $f_i, g_i, h_i$ for simplicity. When the context is clear, we abuse the notation $r$ to represent arbitrary $f$, $g$ or $h$. The corresponding predicting probability of feature $r_i$ is denoted as $p(r_i)$. To summary, there are $b_{m1} + b_{m2}$ features in uni-modal ensemble, $k_{m1} + k_{m2} + k_{pa}$ features in multi-modal training approaches.

We first prove statement (a.), which claims that the number of features learned in multi-modal training approaches are provably less than any of the number of features learned in uni-modal training. The proof depends on the following Lemma B.1.

**Lemma B.1.** *Assume there exists $T$ features $r_i, i = 1, \ldots, T$. If we replace one of the $T$ features (without loss of generality, $r_T$) with a more powerful feature $r'$, where $p(r') > p(r_T)$, then the predicting probability for each data point increases (where the probability is taken over the randomness of the training data).*

We next provide the proof of statements (a.): based on Lemma B.1. We shall prove $k_{m1} + k_{m2} + k_{pa} < b_{m1}$ without loss of generality. Start from the features $f_1(x^{m_1}), \ldots, f_{k_{m1}}(x^{m_1})$ which are common features in both multi-modal training approaches and Uni-modal training. Next step, we add feature $f_{k_{m1}+1}$ in uni-modal approachesand $g_1$ in multi-modal training approaches. Obviously, $p(g_1) > p(f_{k_{m1}+1})$ due to the training priority (or multi-modal training approaches should learn $f_{k_{m1}+1}$ instead of $g_1$). Therefore, the predicting probability of multi-modal training approaches is larger than uni-modal approaches.

Repeating the procedure by comparing $g_i$ with $f_{k_{m1}+i}$ and comparing $h_j$ with $f_{k_{m1}+k_{m2}+j}$, the predicting probability of multi-modal training approaches is always larger than uni-modal approaches. Note that $b_{m1}$ should be always larger than $k_{m1} + k_{m2}$, or the predicting probability of uni-modal

---

[14]The probability is taken over the randomness of the testing point

approaches would be smaller than multi-modal training approaches. At the end of the comparison, the predicting probability of multi-modal training approaches is still larger than uni-modal approaches. This requires that uni-modal approaches should learn more features, which can be regarded as uni-modal approacheslearns a features while multi-modal training approaches learns an empty feature. In conclusion, uni-modal approaches learns more features compared to multi-modal training approaches, leading to $b_{m1} > k_{m1} + k_{m2} + k_{pa}$.

We next prove the statement (b.). The proof of (b.) is based on (a.). We next only consider modality $x^{m_1}$, the proof for modality $x^{m_2}$ is similar. Note that the since the number of features learned in multi-modal training approaches is less than $b_{m1}$, the number of features learned in $x^{m_1}$ must be less than $b_{m1}$ (Note that those features can be either paired feature or uni-modal feature, namely, $f_1, \ldots, f_{k_{m1}}$ and $h_1, \ldots, h_{k_{pa}}$). Therefore, multi-modal training approaches learns less features compared to uni-modal approaches in modality $x^{m_1}$. On the other hand, the predicting probability of features learned in multi-modal training approaches ($f_1, \ldots, f_{k_{m1}}$ and $h_1, \ldots, h_{k_{pa}}$, considering only modality $x^{m_1}$ for the paired feature) is less than that learned in uni-modal approaches ($f_1, \ldots, f_{b_{m1}}$), because otherwise, uni-modal approaches will learn the features in $h$ instead of $f$. In conclusion, when considering only modality $x^{m_1}$, the number of features learned in multi-modal training approaches is less and its corresponding predicting probability is small. Therefore, each modality in multi-modal training approaches performs worse than uni-modal approaches.

We finally prove the statement (c.). Recall that the loss is $-\sum_i u(r_i)$ where $u(r_i) = \mathbb{I}(yr_i > 0) - \mathbb{I}(yr_i < 0)$. Note that $\mathbb{E}(u(r_i)) = \frac{1}{2}p(r_i)$ and $|u(r_i)| \leq 1$. We derive that:

$$\mathbb{P}\left(-\sum_{i \in [k_{m1}]} u(f_i) - \sum_{i \in [k_{m2}]} u(g_i) - \sum_{i \in [k_{pa}]} u(h_i) \leq -\sum_{i \in [b_{m1}]} u(f_i) - \sum_{i \in [b_{m2}]} u(g_i)\right)$$

$$=\mathbb{P}\left(\sum_{k_{m1} < i \leq b_{m1}} u(f_i) + \sum_{k_{m2} < i \leq b_{m2}} u(g_i) - \sum_{i \in [k_{pa}]} u(h_i) \leq 0\right)$$

$$=\mathbb{P}\left(\sum_{k_{m1} < i \leq b_{m1}} u(f_i) + \sum_{k_{m2} < i \leq b_{m2}} u(g_i) - \sum_{i \in [k_{pa}]} u(h_i) + \frac{1}{2}E \leq \frac{1}{2}E\right),$$

where $E = -\mathbb{E}(\sum_{k_{m1} < i \leq b_{m1}} u(f_i) + \sum_{k_{m2} < i \leq b_{m2}} u(g_i) - \sum_{i \in [k_{pa}]} u(h_i)) = \sum_{i \in [k_{pa}]} p(h_i) - \sum_{k_{m1} < i \leq b_{m1}} p(f_i) - \sum_{k_{m2} < i \leq b_{m2}} p(g_i)$. Due to the training priority and the conclusion in (a.),

$$\sum_{i \in [b_{m1}+1, b_{m1}+b_{m2}]} p_{[i]} \leq \sum_{k_{m1} < i \leq b_{m1}} p(f_i) + \sum_{k_{m2} < i \leq b_{m2}} p(g_i).$$

Therefore, $E \leq \sum_{i \in [k_{pa}]} p(h_i) - \sum_{i \in [b_{m1}+1, b_{m1}+b_{m2}]} p_{[i]} \leq \sqrt{8(k_{pa} + b_{m1} - k_{m1} + b_{m2} - k_{m2}) \log(1/\delta)}$. We next apply Hoeffding inequality on Equation B.1 and derive that

$$\mathbb{P}\left(-\sum_{i \in [k_{m1}]} u(f_i) - \sum_{i \in [k_{m2}]} u(g_i) - \sum_{i \in [k_{pa}]} u(h_i) < -\sum_{i \in [b_{m1}]} u(f_i) - \sum_{i \in [b_{m2}]} u(g_i)\right)$$
$$\leq \exp(-E^2/8(k_{pa} + b_{m1} - k_{m1} + b_{m2} - k_{m2}))$$
$$\leq \delta$$

To conclude, multi-modal training approaches outperform uni-modal ensemble concerning the testing loss with probability at least $1 - \delta$.

Compared to uni-modal ensemble, denote the additional paired feature are indexed by $c$, and the additional uni-modal feature in uni-modal ensemble are indexed by $v$. We have that:

$$\mathbb{P}(\sum_{i\in[c]}(I(f_i(x)>0)-I(f_i(x)<0))-\sum_{j\in[v]}(I(f_j(x)>0)-I(f_j(x)<0))>0)$$

$$=\mathbb{P}(\sum_{i\in[c]}I(f_i(x)>0)-\sum_{j\in[v]}I(f_j(x)>0)-\frac{1}{2}[\sum_{i\in[c]}p_i-\sum_{j\in[v]}p_j]>\frac{1}{2}[\sum_{j\in[v]}p_j-\sum_{i\in[c]}p_i]) \quad (1)$$

$$\leq\exp(-(\sum_{j\in[v]}p_j-\sum_{i\in[c]}p_i)^2/8|c+v|)$$

Therefore, if $\sum_{j\in[v]}p_j-\sum_{i\in[c]}p_i\geq\sqrt{8(c+v)\log(1/\delta)}$, the probability is done. Therefore, for a new data point, uni-modal ensemble can outperforms multi-modal training approaches with high probability.

$\square$

***Proof of Lemma B.1.*** We define $r_{[-T]}$ as the features $r_1,\ldots,r_{T-1}$. The proof is divided into two parts, depending on whether $\sum_{i\in[T-1]}\mathbb{I}(r_i\neq 0)$ is even or odd. We regard the term $\sum_{i\in[T-1]}\mathbb{I}(r_i\neq 0)$ as the number of effective features in $r_{[-T]}$. To simplify the discussion, we rescale $r$ such that $|yr|=1$ (when $r\neq 0$) or $|yr|=0$ (when $r=0$).

*Case 1*: When the number of effective features in $r_{[-T]}$ is even. (a. ) If $|\sum_{i\in[T-1]}yr_i|\geq 2$, adding $r_T$ or $r'$ does not alter the predicting probability, namely

$$\mathbb{P}\left(y\left[r_T+\sum_{i\in[T-1]}yr_i\right]>0\ \Big|\ \Big|\sum_{i\in[T-1]}yr_i\Big|\geq 2\right)+\frac{1}{2}\mathbb{P}\left(y\left[r_T+\sum_{i\in[T-1]}yr_i\right]=0\ \Big|\ \Big|\sum_{i\in[T-1]}yr_i\Big|\geq 2\right)$$

$$=\mathbb{P}\left(y\left[r'+\sum_{i\in[T-1]}yr_i\right]>0\ \Big|\ \Big|\sum_{i\in[T-1]}yr_i\Big|\geq 2\right)+\frac{1}{2}\mathbb{P}\left(y\left[r'+\sum_{i\in[T-1]}yr_i\right]=0\ \Big|\ \Big|\sum_{i\in[T-1]}yr_i\Big|\geq 2\right).$$

(b. ) When the number of effective features in $r_{[-T]}$ is even, $|\sum_{i\in[T-1]}yr_i|\neq 1$.

(c. ) When $|\sum_{i\in[T-1]}yr_i|=0$, due to the assumption that $p(r')>p(r_T)$ and $\epsilon(r)=p(r)/c$, adding $r'$ helps increase the predicting probability compared to $r_T$, namely

$$\mathbb{P}\left(y\left[r_T+\sum_{i\in[T-1]}yr_i\right]>0\ \Big|\ \Big|\sum_{i\in[T-1]}yr_i\Big|=0\right)+\frac{1}{2}\mathbb{P}\left(y\left[r_T+\sum_{i\in[T-1]}yr_i\right]=0\ \Big|\ \Big|\sum_{i\in[T-1]}yr_i\Big|=0\right)$$

$$>\mathbb{P}\left(y\left[r'+\sum_{i\in[T-1]}yr_i\right]>0\ \Big|\ \Big|\sum_{i\in[T-1]}yr_i\Big|=0\right)+\frac{1}{2}\mathbb{P}\left(y\left[r'+\sum_{i\in[T-1]}yr_i\right]=0\ \Big|\ \Big|\sum_{i\in[T-1]}yr_i\Big|=0\right).$$

The above inequality is derived based on the following equation:

$$\mathbb{P}\left(y\left[r_T+\sum_{i\in[T-1]}yr_i\right]>0\ \Big|\ \Big|\sum_{i\in[T-1]}yr_i\Big|=0\right)+\frac{1}{2}\mathbb{P}\left(y\left[r_T+\sum_{i\in[T-1]}yr_i\right]=0\ \Big|\ \Big|\sum_{i\in[T-1]}yr_i\Big|=0\right)$$

$$=\mathbb{P}\left(yr_T>0\ \Big|\ \Big|\sum_{i\in[T-1]}yr_i\Big|=0\right)+\frac{1}{2}\mathbb{P}\left(yr_T=0\ \Big|\ \Big|\sum_{i\in[T-1]}yr_i\Big|=0\right)$$

$$=p(r_T)+\frac{1}{2}\left[1-p(r_T)-\epsilon(r_T)\right]$$

$$=\frac{1}{2}\left[1+(1-1/c)p(r_T)\right].$$

Since we assume $c>1$, the probability increases with probability $p(r_T)$.

Therefore, under the three conditions, adding $r'$ increase the predicting probability more compared to $r_T$. In summary, under case 1 (a-c), adding $r'$ increase the predicting probability compared to $r_T$.

*Case 2*: When the number of features in $r_{[-T]}$ is odd. The discussion in (b.) can be a little bit more complex compared to case 1.

(a. ) If $|\sum_{i\in[T-1]} yr_i| \geq 2$, similar to case 1, adding $r_T$ or $r'$ does not alter the predicting probability, namely

$$
\mathbb{P}\left(y\left[r_T + \sum_{i\in[T-1]} yr_i\right] > 0 \,\bigg|\, \left|\sum_{i\in[T-1]} yr_i\right| \geq 2\right) + \frac{1}{2}\mathbb{P}\left(y\left[r_T + \sum_{i\in[T-1]} yr_i\right] = 0 \,\bigg|\, \left|\sum_{i\in[T-1]} yr_i\right| \geq 2\right)
$$
$$
=\mathbb{P}\left(y\left[r' + \sum_{i\in[T-1]} yr_i\right] > 0 \,\bigg|\, \left|\sum_{i\in[T-1]} yr_i\right| \geq 2\right) + \frac{1}{2}\mathbb{P}\left(y\left[r' + \sum_{i\in[T-1]} yr_i\right] = 0 \,\bigg|\, \left|\sum_{i\in[T-1]} yr_i\right| \geq 2\right).
$$

(b. ) If $|\sum_{i\in[T-1]} yr_i| = 1$: (b.1 ) If $\sum_{i\in[T-1]} yr_i = -1$:

$$
\mathbb{P}\left(y\left[r_T + \sum_{i\in[T-1]} yr_i\right] > 0 \,\bigg|\, \sum_{i\in[T-1]} yr_i = -1\right) + \frac{1}{2}\mathbb{P}\left(y\left[r_T + \sum_{i\in[T-1]} yr_i\right] = 0 \,\bigg|\, \sum_{i\in[T-1]} yr_i = -1\right)
$$
$$
=\mathbb{P}\left(yr_T - 1 > 0 \,\bigg|\, \sum_{i\in[T-1]} yr_i = -1\right) + \frac{1}{2}\mathbb{P}\left(yr_T - 1 = 0 \,\bigg|\, \sum_{i\in[T-1]} yr_i = -1\right)
$$
$$
=\frac{1}{2}\mathbb{P}\left(yr_T - 1 = 0 \,\bigg|\, \sum_{i\in[T-1]} yr_i = -1\right)
$$
$$
=\frac{1}{2}p(r_T).
$$

(b.2 ) If $\sum_{i\in[T-1]} yr_i = +1$:

$$
\mathbb{P}\left(y\left[r_T + \sum_{i\in[T-1]} yr_i\right] > 0 \,\bigg|\, \sum_{i\in[T-1]} yr_i = 1\right) + \frac{1}{2}\mathbb{P}\left(y\left[r_T + \sum_{i\in[T-1]} yr_i\right] = 0 \,\bigg|\, \sum_{i\in[T-1]} yr_i = 1\right)
$$
$$
=\mathbb{P}\left(yr_T + 1 > 0 \,\bigg|\, \sum_{i\in[T-1]} yr_i = 1\right) + \frac{1}{2}\mathbb{P}\left(yr_T + 1 = 0 \,\bigg|\, \sum_{i\in[T-1]} yr_i = 1\right)
$$
$$
=(1 - \epsilon(r_T)) + \frac{1}{2}\epsilon(r_T)
$$
$$
=1 - \frac{1}{2c}p(r_T).
$$

Note that the probability of event (b.1) and the probability of event (b.2) satisfy the following equation by Lemma B.2:

$$
\mathbb{P}\left(\sum_{i\in[T-1]} yr_i = 1\right) = c\mathbb{P}\left(\sum_{i\in[T-1]} yr_i = -1\right). \tag{2}
$$

Therefore, the total probability under case (b) is

$$
\frac{1}{2}p(r_T)\mathbb{P}\left(\sum_{i\in[T-1]} yr_i = -1\right) + (1 - \frac{1}{2c}p(r_T))\mathbb{P}\left(\sum_{i\in[T-1]} yr_i = 1\right)
$$
$$
=\mathbb{P}\left(\sum_{i\in[T-1]} yr_i = 1\right)
$$

which is independent of $p(r_T)$. Therefore, adding $r_T$ or $r'$ share the same predicting probability.

(c. ) When the number of effective features in $r_{[-T]}$ is odd, $|\sum_{i\in[T-1]} yr_i| \neq 0$.

In summary, under case 2 (a-c), adding $r'$ do not decrease the predicting probability compared to $r_T$.

The following lemmas are used during the proof.

**Lemma B.2.** *Consider $T-1$ features $r_1, \ldots, r_{T-1}$, the following equation holds:*

$$\mathbb{P}\left(\sum_{i \in [T-1]} yr_i = 1\right) = c\mathbb{P}\left(\sum_{i \in [T-1]} yr_i = -1\right). \tag{3}$$

*Proof.* It can be proved to compare the events $A = \{\sum_{i \in [T-1]} yr_i = 1\}$ and $B = \{\sum_{i \in [T-1]} yr_i = -1\}$. Every event in $A$ has a complementary event in $B$, namely,

$$yr_i = 1 \text{ in } B \text{ if } yr_i = -1 \text{ in } A$$
$$yr_i = -1 \text{ in } B \text{ if } yr_i = 1 \text{ in } A$$
$$yr_i = 0 \text{ in } B \text{ if } yr_i = 0 \text{ in } A$$

Comparing each event in $A$ with its complementary event in $B$ leads to the conclusion. $\square$

Combining case 1 and case 2 together leads to the final conclusion.

$\square$

## B.2 GENERALIZE THEOREM 3.4 TO MORE MODALITIES

We next show that the results in Theorem 3.4 can be generalized to the regime of more modals.

Specifically, we assume a $T$-modal regimes, and denote the modals as $x^{m_i}, i \in [T]$. In uni-modal pre-training approaches, let $b_{m_i}$ denote the number of returned features in modal $i$. In multi-modal joint training, let $k_{m_i}$ denote the number of uni-modal features for modal $i$, and $k_{pa}$ denote the number of returned paired features. We derive the following Theorem B.3 for the multi-modal regimes.

**Theorem B.3.** *Based on the above notations, we provide three types of laziness from three perspectives:*

(a. ) **Quantity Laziness**: $\sum_i k_{m_i} + k_{pa} \leq \min_i\{b_{m_i}\}$.

(b. ) **Uni-modal Laziness**: *Each modality in multi-modal training approaches performs worse than uni-modal training.*

(c. ) **Performance Laziness**: *Consider a new testing point, then for every $\delta > 0$, if the following inequality holds:*

$$\sum_{i \in [k_{pa}]} p(h_i) \leq \sum_{i \in [\min_i\{b_{m_i}\}+1, \sum_i b_{m_i}]} p_{[i]} + \Delta(\delta),$$

*where $\Delta(\delta) = \sqrt{8(k_{pa} + \sum_j [b_{m_j} - k_{m_j}]) \log(1/\delta)}$, then with probability[15] at least $1 - \delta$, uni-modal ensemble outperform multi-modal training approaches concerning the loss on the testing point with probability.*

## B.3 PROOF OF THEOREM 3.5

**Theorem 3.5.** *Denote the paired features by $h_1, \ldots h_L$ with corresponding predicting probability $p(h_1), \ldots, p(h_L)$. Assume that distillation can boost the training priority by $p^0 > 0$. If there exists paired features whose predicting probability exceeds the boosting probability $p^0$, namely, the set $\mathcal{S}$ is not empty:*

$$\mathcal{S} = \{h_i : p(h_i) > p^0\} \neq \phi.$$

*Then UMT helps uni-modal feature learning and can also learn easy-to-learn paired features.*

---

[15]The probability is taken over the randomness of the testing point

Table 23: Dataset used in Example B.4. $+$ means the feature is larger than zero and $-$ means the feature is less than zero. We denote the predicting probability by $p$ and the rectified probability (due to pushing force) by $p'$.

| | $f_1$ | $f_2$ | $f_3$ | $g_1$ | $g_2$ | $g_3$ | $h$ | y |
|---|---|---|---|---|---|---|---|---|
| $p$ | 0.20 | 0.10 | 0.05 | 0.15 | 0.08 | 0.02 | 0.28 | / |
| $p'$ | 0.35($\uparrow$) | 0.25($\uparrow$) | 0.20($\uparrow$) | 0.32($\uparrow$) | 0.23($\uparrow$) | 0.17($\uparrow$) | 0.28 | / |
| data a | + | + | + | + | - | + | + | +1 |
| data b | 0 | + | 0 | + | + | - | + | +1 |
| data c | + | + | 0 | - | + | + | 0 | -1 |
| data d | + | - | + | + | + | 0 | + | -1 |

*Proof.* The core of Theorem 3.5 is to clarify the training priority. We revisit the notations of Theorem 3.4 as follows without further clarification. At the end of the training, uni-modal ensemble learn $b_{m1} + b_{m2}$ useful features, namely, $f_1, \ldots, f_{b_{m1}}, g_1, \ldots, g_{b_{m2}}$. And multi-modal training approaches learn $k_{m1} + k_{m2} + k_{pa}$ features: $f_1, \ldots, f_{k_{m1}}, g_1, \ldots, g_{k_{m2}}, h_1, \ldots, h_{k_{pa}}$. We note that there are still many empty features $e_i$ in the model due to the initialization.

By distillation, the model learns the features according to the new priority. Since the set $\mathcal{S}$ is not empty, there exists paired features that is learned before the empty features. By distillation, the model would learn all the useful features that appear in uni-modal approaches, as well as those features in set $\mathcal{S}$. Therefore, UMT outperforms uni-modal ensemblewhen there exists useful paired features.

$\square$

## B.4 A CONCRETE EXAMPLE TO ILLUSTRATE THEOREM 3.4

We next provide a concrete example to better illustrate the Modality Laziness issues. For Example B.4, we aim to show the Modality Laziness issues. For Example B.5, we aim to show the role of the pushing force.

*Example* B.4. Consider modality $x^{m_1}$ with features $f_1, f_2, f_3$ (corresponding prediction probability $p = 0.2, 0.1, 0.05$), and modality $x^{m_2}$ with features $g_1, g_2, g_3$ (corresponding prediction probability $p = 0.15, 0.08, 0.02$). We show the dataset in Table 23 and aim to minimize the training loss to zero.

In uni-modal approaches, we learn features $f_1$, $f_2$ and $f_3$ on modality $x^{m_1}$ (similarly, $g_1$, $g_2$, and $g_3$ on modality $x^{m_2}$). Therefore, we learn features $f_1, f_2, f_3, g_1, g_2, g_3$ in uni-modal ensemble. In multi-modal training approaches without paired feature, we can only learn three features $f_1, f_2, g_2$ due to the training priority $f_1 > g_1 > f_2 > g_2 > f_3 > g_3$ (decreasing order in $p$). This phenomenon is caused by modality laziness.

We next consider another paired feature $h$ with probability $p = 0.28$. Under the case, multi-modal training approaches only learn two features $h$ and $f_1$. Therefore, when $h$ is not powerful enough, uni-modal ensemble outperforms multi-modal training approaches.

*Example* B.5. We follow the notations and dataset in Example B.4. By applying the pushing force, assume that each probability of uni-modal feature boosts 0.15, which changes the training priority to $f_1 > g_1 > h > f_2 > g_2 > f_3 > g_3$ (decreasing order in $p'$). Therefore, multi-modal training approaches (with pushing force) learns $f_1, f_2, h$. As a comparison, multi-modal training approaches (without pushing force) can only learn $f_1, h$. Therefore, pushing force helps learn more features. We additionally remark that we only consider the training error in this example, and there might be other penalties in practice (*e.g.*, distillation loss).

