# OpenReview forum: "On Uni-modal Feature Learning in Multi-modal Learning"
_ICLR.cc/2023/Conference — Submitted to ICLR 2023_

### Official Review · Reviewer_fVP6 · 2022-10-20

**Confidence:** 4
**Clarity, Quality, Novelty And Reproducibility:** The writing is fine, and the descript…
**Correctness:** 3
**Technical Novelty And Significance:** 3
**Empirical Novelty And Significance:** 2
**Recommendation:** 5

**Strength And Weaknesses:**

S1) a good motivation for the features yielded from multi-modal fusion modules and the motivation is shown experimentally.
S2) Mathematical description for the generalization degradation

W1) the detail of the uni-modal model is missed in UMT. If the uni-modal model is such a large network, UMT has benefits in terms of generalization.
W2) Fewer baselines. Are there no works or techniques to preserve uni-modal information?
W3) The author insists that the existing fusion modules delete the important uni-modal information somewhat. In UMT, the knowledge distillation is performed before the fusion module. Following the author's insistence, the distilled knowledge can also be corrupted after fusion. What is the reason for the design choice?
W4) To select UMT or UME, the testing performance is required. It makes the proposed methods less mature. Is there any automatic rule for this?


**Summary Of The Paper:**

This paper tackled the modality laziness in current multi-modal fusion methods and suggested leveraging the uni-modal feature in two ways. Depending on the importance of paired features (maybe multi-modal information), simple averaging uni-modal features or knowledge distillation from strong uni-modal models. Also, the generalization degradation is described in theoretical perspective.

**Summary Of The Review:**

Please handle the weakness in the author's response

---

> ### Author Response · Authors · 2022-11-06
> **Thanks and response to concerns (Part I).**
>
> Thanks for your review, below we address your concerns.
>
> > **1, the detail of the uni-modal model**
>
> Thanks for reminding.
>
> The uni-modal model we used is not a larger network.
>
> For a specified modality, the same backbone is used as the feature extractor in the uni-modal model and multi-modal model.
>
> For example, if we use ResNet18 in uni-modal training, we also use the backbone of ResNet18 to extract the features of the corresponding modality in the multi-modal model.
>
> _In the revised version of our paper, we have made this detail clear._
>
> > **2,  Fewer baselines. Are there no works or techniques to preserve uni-modal information?**
>
> Although Modality Laziness is an urgent problem to be solved, there are **not** many effective baselines at present.
>
> _2.1 In late-fusion learning, the main comparing methods are Gradient Blending[1] and OGM-GE[2]._
>
> In [1], all the methods compared with Gradient Blending are not specifically designed to preserve uni-modal information. So the baselines show low performance. Only Gradient Blending is designed to preserve uni-modal information and gets good performance.  In our paper, we compare our method with G-Blending and outperform it.
>
> In [2], its authors mainly compared OGM-GE with Gradient Blending.  Although they compare OGM-GE with FiLM, TSN-AV, TSM-AV, TBN, PSP, and so on, these methods are not proposed to solve the unbalance training problem in multi-modal training, so they did not show considerable performance. In our paper, we have compared our method with OGM-GE, and OGM-GE has outperformed those methods, and we outperform OGM-GE.
>
>
> _2.2 In intermediate-fusion learning, the main comparing method is Balanced Multi-modal Algorithm[3]._
>
> [3] proposes Balanced Multi-modal Algorithm for intermediate-fusion learning. However, they only compare their method with RUBi[4], which is a method proposed for Visual Question Answering, but not for their tasks. So RUBi does not show good performance. Our method outperforms RUBi and Balanced Multi-modal Algorithm on the dataset that [3] used.
>
> Both [1] and [3] are **recent**(published in this year) papers which focus on mitigating imbalance multi-modal training issues.
>
> > **3 In UMT, the knowledge distillation is performed before the fusion module. Following the author's insistence, the distilled knowledge can also be corrupted after fusion**
>
> Your understanding of UMT is right. That "the distilled knowledge can also be corrupted after fusion" is why Uni-Modal Ensemble(UME) was proposed.
>
> _3.1 Consider a multi-modal dataset with many uni-modal features but few paired features,_ fusion module not only affects the learning of uni-modal features, but also has little potential positive effect.
> In view of this situation, we propose to use UME. Combining predictions of uni-modal models avoids insufficient learning of uni-modal features by nature. And UME does perform well on UCF101 and ModelNet40.
>
> _3.2 On the other hand, when the multi-modal dataset contains some paired features_, the fusion module can play its role to learn paired features. That is why we retain the fusion module in UMT. Besides, the uni-modal distillation of UMT at the feature level does work and it alleviates the Modality Laziness problem a lot. Combining these two points, UMT can achieve very good performance on VGG-Sound and Kinetics-400.
>
> > **4, About the practicality of the method and automatic rule?**
>
> _4.1 About the practicality of the method_
>
> One of the main comparing methods, Gradient Blending[1], needs another split training dataset and the validation set to estimate the overfitting-to-generalization ratio from the training accuracy and the validation accuracy to re-weight the losses and then re-train the model again and again.
>
> Although OGM-GE[2] is one-stage, however, it needs to tune too many hyper-parameters, including the start and end epoch of the gradient modulation, an “alpha” used to calculate the coefficients for the modulation and whether adaptive Gaussian noise Enhancement (GE) is needed. The more complicated thing is that these hyper-parameters need to be re-tuned on new datasets. **Finding the right hyper-parameters for OGM-GE in a new task is far more troublesome than our method**
>
> Currently, existing one-stage method[2] is not only complicated to implement but also worse than our method.
> Although our method needs to know the testing accuracy or the Val accuracy to choose the proper algorithm from UME and UMT, it's not only simple to understand the logic and reimplement on any new multi-modal task, but also effective, which outperforms other methods on various datasets.
>
> As reviewer DNPE said about our method, "The proposed solution is simple but effective". Compared with the previous methods, our method is very easy to reproduce and very practical.

---

> > ### Author Response · Authors · 2022-11-07
> > **Thanks and response to your concerns (Part II)**
> >
> > _4.2 Automatic rule_
> >
> >
> > Different multimodal tasks have very different characteristics, so there are various training and fusion methods. In a recent multimodal analysis paper[5], totally different multimodal models are used in six multimodal datasets.
> >
> > In VQA, as discussed in Sec 3.1 of our paper, "the same image with different text questions may have totally different labels, making it pointless to check its uni-modal accuracy." Uni-modal accuracy is meaningless in tasks like that, and effective information exchange is the most important.
> >
> > Also, advanced joint training methods are not as practical as Uni-Modal Ensemble(UME) in datasets where uni-modal features are totally dominant, such as ModelNet40 or UCF101.
> >
> > And in datasets like VGG-Sound, both uni-modal features and paired features are essential, and we need both of them. UMT can show its talent in this situation.
> >
> > Our paper not only proposes UMT and UME but also classifies the features of multimodal datasets and gives practical guidance for multimodal training: we should select effective algorithms according to the proportions of uni-modal and paired features. Based on our theoretical analysis, we think it is challenging to have an algorithm that can perform well on all multimodal tasks.
> >
> > And our paper tries to classify different multimodal tasks and proposes that we should design different multimodal algorithms according to the characteristics of multimodal datasets. We consider this to be one of the most important contributions of our paper.
> >
> > **We think we need a step to classify different multimodal tasks, which is necessary. And our trick is simple enough.**
> >
> > "The proposed solution is simple, and easy to reproduce." as Reviewer DNPE said.
> >
> > **Reference**
> >
> > [1] What Makes Training Multi-Modal Classification Networks Hard? CVPR'20
> >
> > [2] Balanced Multimodal Learning via On-the-fly Gradient Modulation CVPR'22 oral
> >
> > [3] Characterizing and overcoming the greedy nature of learning in multi-modal deep neural networks. ICML'22
> >
> > [4] Rubi: Reducing unimodal biases in visual question answering. NeurIPS'19
> >
> > [5] MultiViz: An Analysis Benchmark for Visualizing and Understanding Multimodal Models

---

> > > ### Author Response · Authors · 2022-11-07
> > > **Rebuttal Revision**
> > >
> > > Thanks for your review again, and we have updated our paper to clarify the confusing part:
> > >
> > > **1,** We have updated the part about UMT in Sec 3.3 and pointed out that for a specified modality, the same backbone is used as the feature extractor in the uni-modal model and multi-modal model.
> > >
> > > **All modified texts are marked in red**
> > >
> > >
> > > If you have any other questions, we are happy to provide additional discussion to allay your concerns.

---

> > > > ### Author Response · Authors · 2022-11-15
> > > > **Follow-up**
> > > >
> > > > Thank you again for your constructive comments and suggestions. If we have successfully addressed your questions, we would strongly appreciate an increased score. Otherwise, please let us know and we are happy to provide additional experiments and/or discussion to allay your concerns.

---

> > > > > ### Author Response · Authors · 2022-12-01
> > > > > **Do our responses address your concerns?**
> > > > >
> > > > > Dear Reviewer fVP6:
> > > > >
> > > > > Thank you again for your constructive comments and suggestions.
> > > > >
> > > > > If we have successfully addressed your questions, we would strongly appreciate an increased score.
> > > > > Otherwise, please let us know and we are happy to provide additional experiments and/or discussion to allay your concerns.
> > > > >
> > > > > Regards,
> > > > >
> > > > > The authors.

---

> ### Author Response · Authors · 2022-12-07
> **Did our answer address your concerns?**
>
> Dear Reviewer:
>
> Thank you for taking the time to review our paper and providing constructive comments.
>
> We have provided very detailed answers to your concerns and updated the paper. It has been almost a month since we replied to you, and we sincerely want to know whether our reply has successfully addressed your concerns.
>
> If we address your concerns, we would strongly appreciate an increased score. Otherwise, please let us know and we are happy to provide additional experiments and/or discussion to allay your concerns.
>
> Regards,
>
> The authors.

---

### Official Review · Reviewer_AaMz · 2022-10-25

**Confidence:** 4
**Clarity, Quality, Novelty And Reproducibility:** See Strong&Weak
**Correctness:** 3
**Technical Novelty And Significance:** 2
**Empirical Novelty And Significance:** 3
**Recommendation:** 6

**Strength And Weaknesses:**

Strong & Weak:
S1: The writing is easy to follow. A reproducibility code is given. Better experimental results are achieved than the complex late-fusion models.
S2: The architecture of Uni-Modal Teacher is straightforward and is empirically better than classic distillation methods including soft-label.

W1: Although this paper cares about the weakness of multimodal learning and focuses on the importance of unimodal feature learning, assuming that unimodal priors are important and meaningful like previous works (Wu et al, 2022; Peng et al, 2022), it may be better to go further to design a unified, principal framework which can also subsume the settings where unimodal priors are not essential, for example on the visual question answer task. In other words, such a unified framework is agnostic to the unimodal priors and works well for both essential priors (including video classification and action recognition tasks) and meaningless priors (including visual question answer task). Perfectly, one does not need to choose which style of multimodal learning should be adopted to solve their own interest tasks at hand.


**Summary Of The Paper:**

Summary:
This paper proposes to care about the learning of unimodal features in the setting of multimodal fusion. They find out that one modality may strongly dominate the multimodal learning than other modalities on some tasks, like video classification and action recognition. This observation is consistent of previous related works, like greedy nature of multimodal DNN (Wu et al, 2022), under-optimized unimodal representations caused by strong dominated modality (Peng et al, 2022). They argue that previous works are complex to implement and are empirically inferior on some evaluation datasets (UCF101 and ModelNet40). Instead, this paper introduces Uni-Modal Teacher (UMT) to distill pretrained
unimodal features to help learn corresponding unimodal features counterparts in the multimodal model. Furthermore, a trick is used to decide when to use the joint multimodal training (Uni-Modal Teacher, UMT) or just use the simple averaging predictions of unimodal models alone (Uni-Modal Ensemble, UME).


**Summary Of The Review:**

See Strong&Weak

---

> ### Author Response · Authors · 2022-11-07
> **Thanks and Some Discussion**
>
> **Thanks for your nice review. Below we provide some discussion about the “unified, principal framework for multi-modal learning”.**
>
> Recently, some works on general artificial intelligence have emerged. Almost all the models in these papers are transformers[1,2,3].
>
> We believe that such a unified model needs to satisfy three conditions:
>
> (1) This model can handle all kinds of data. The transformer is a good candidate, which can tokenize everything.
>
> (2) The model should be large enough. Larger models have a greater capacity to learn more.
>
> (3) The model should be pre-trained on various and large-scale datasets. Otherwise, it's easily overfitting downstream tasks.
>
> Gato[1] demonstrates how to pre-train a multi-task and multi-modal large model. They tokenize everything and predict the masked inputs.  After the pre-training, they can perform well on various downstream tasks. However, Gato still doesn't cover all modalities, such as audio and optical flow.
> UNIFIED-IO[2] and BEiTv3[3] focus on the image and text modalities, and there is also room for improvement.
>
> However, training any of these models requires too many computing resources, which the vast majority of machine learning labs cannot afford.
>
> Therefore, we believe that there are two important research directions:
>
> (1) Train a large model with limited computing resources
>
> (2) Train a large model with more modalities.
>
> Thanks for your review again, and hope the discussion is useful.
>
> **Reference**
>
> [1] A Generalist Agent. Deepmind
>
> [2] BEiT v3: Image as a foreign language: Beit pretraining for all vision and vision-language tasks
>
> [3] Unified-IO: A Unified Model for Vision, Language, and Multi-Modal Tasks

---

### Official Review · Reviewer_DNPE · 2022-10-26

**Confidence:** 3
**Correctness:** 3
**Technical Novelty And Significance:** 4
**Empirical Novelty And Significance:** 3
**Recommendation:** 8

**Clarity, Quality, Novelty And Reproducibility:**

The paper is well-motivated and mostly well presented. It addresses an interesting phenomenon in supervised multimodal learning and provides a theoretical analysis. The proposed solution is simple, and easy to reproduce.

**Strength And Weaknesses:**

Strength:

-The paper discovers an interesting phenomenon of Modality Laziness, and provides an in-depth understanding in both empirical and theoretical aspects.

-The paper is well motivated. The analytical experiment in Table 1 really shows the severeness of existing supervised multimodal learning.

-I appreciate the efforts that the authors try to explain the key concepts in an intuitive way. For example, the figure 3 is helpful.

-The results compared with several existing methods are competitive. The proposed solution is simple but effective.



Weakness

-There is one method called “bimodal deep autoencoder” [Ngiam et al.], which can also be applied for joint feature extraction, then used for supervised learning. The discussion along this line is missing. Since  the motivation of this work is address Modality Laziness, the auto-encoder used in [Ngiam et al.] also has the capability to mitigate the insufficient learning. A comparison with this work is strongly recommended.

[Ngiam et al,] Jiquan Ngiam et al., Multimodal Deep Learning, ICML 2011

-The term of “naive multimodal training” appears multiple times across the paper, and its suffers from Modality Laziness. While the formal definition of this term is unclear to me.

**Summary Of The Paper:**

This paper studies the insufficiency phenomenon of uni-modal feature learning in supervised multi-modal learning problem. The authors identify that the recent late-fusion method suffers from insufficient learning issue, called as Modality Laziness. To understand this phenomenon, the authors conduct a series of experiments and make a theoretical analysis. The authors propose two late fusion learning methods, Uni-Model Ensemble and Uni-Model Teacher, which work better in different cases according to the distribution of uni-modal and paired features. A simple guiding strategy is also provided. Experiments show the advantages of the proposed method in several multimodal datasets.

--- post-rebuttal ---\
Thanks for the authors' response. After reading the response and other reviewers' comments, my concerns are addressed. I keep my score.

**Summary Of The Review:**

Overall, I think this work is well executed, with a strong motivation and good theoretical and empirical analysis. In terms of existing work, I'd like to see how this work compares with multimodal deep learning [Ngiam et al.].

---

> ### Author Response · Authors · 2022-11-06
> **Thanks and response to your questions**
>
> Thanks for your review, and we are happy that you think our work is well executed! Below we answer your questions.
>
> **1, The term “naive multimodal training”**
>
> In multi-modal late-fusion learning, each modality is encoded by its corresponding encoder and then a fusion module is applied on top of them to produce outputs.
>
> Naive multi-modal training means multi-modal late-fusion learning without carefully designed methods (carefully designed methods include OGM-GE[1], G-Blending[2], and so on).
>
> Thanks for pointing it out, and we will clarify that in the revised version of the paper.
>
> **2, About bimodal deep autoencoder**
>
> Thanks for providing this related paper.
>
> We believe that proper unsupervised pre-training will definitely help to alleviate the Modality Laziness problem.
>
> Although our paper currently does not cover unsupervised pre-training,
> in Appendix A.6, we use the uni-modal supervised pre-trained encoders' parameters as the initialized weights in multi-modal training and randomly initialize a multi-modal linear classifier on the encoders in VGG-Sound. **_This method does outperform the naive multi-modal training, however, it's still worse than UMT._**
> Because the multi-modal linear layer is randomly initialized and it hurts the encoders in the initial stage of training. So just training the multi-modal model starting from the uni-modal pre-trained weights is not enough.
>
> Thanks for providing this related paper again, and we will add a discussion about it in the next version of the paper.
>
> **Reference**
>
> [1] Balanced Multimodal Learning via On-the-fly Gradient Modulation CVPR'22 oral
>
> [2] What Makes Training Multi-Modal Classification Networks Hard? CVPR'20

---

> > ### Author Response · Authors · 2022-11-07
> > **Rebuttal Revision**
> >
> > Thanks for your review again, and we have updated our paper to clarify the confusing part:
> >
> > **1,**  “Naive Multi-modal Training” first appeared in sec 3.2, and we have marked its meaning.
> >
> > **2,** In Appendix A.6, we add a discussion with [1].
> >
> > **All modified texts are marked in red**
> >
> >
> > **Reference**
> >
> > [1] Jiquan Ngiam et al., Multimodal Deep Learning, ICML 2011

---

### Official Review · Reviewer_5mXz · 2022-10-28

**Confidence:** 4
**Correctness:** 3
**Technical Novelty And Significance:** 3
**Empirical Novelty And Significance:** 3
**Recommendation:** 5

**Clarity, Quality, Novelty And Reproducibility:**

If all of details has no problems, and so all of claims are validated, I think that the idea is novel and the contributions are explicit, and this work is worth publishing only considering the achievements and results.

However, some parts of details are missing as described Weaknesses above.
I think that it needs to re-organize and re-write the manuscript with respect to clarity and reproducibility.


**Strength And Weaknesses:**

*Strength
- To identify interesting conceptualization on uni-modal features and paired features
- To provide experimental evidences (mostly shown in Appendices) to justify their arguments and choices on this work
- To explain theoretical aspects to characterize modality laziness

*Weaknesses
- It seems that some terms in the paper are NOT so clear: what is the exact definition of modality laziness? Why should we utilize new term? What are the discriminating points for it?
- This paper is hard to follow due to lack of explanation for core parts such as model architectures of uni-modal teacher, loss functions with λ_task, λ_distill, λ_distill in Algorithm 1 or the citation of reference works, which also needs for reproducibility.
- The organization of this manuscript makes readers confusing. It seems that theoretical part (Section 3.4) and the others are totally separated. Uni-modal features and paired features also are redefined in Section 3.4. On the other hand, there is no experimental result related to Section 3.4 in main body of the manuscript (but in Appendices).
- Even though related work Section is not bad, it would be better to compare one of the recent works [1], which deeply related to this work.

[1] Modality Competition: What Makes Joint Training of Multi-modal Network Fail in Deep Learning? (Provably), ICML 2022

**Summary Of The Paper:**

This paper deals with multi-modal joint training methods on deep neural networks.
Related to this topic, several works have reported that the best uni-modal networks outperform the multi-modal networks even though the multi-modal networks receive more information.
This work is focused on this topic, and the main contributions of this paper are two-fold: (1) to propose an improved methodology with experimental supports on audio-video data for sound classification (VGG-Sound), audio-video data for action recognition (UCF-101, Kinetics-400), front-rear view of object data (ModelNet40), (2) to investigate theoretical results on characteristics among uni-modal features (learned from uni-modal training), and paired features (only learned from cross-modal interaction).


**Summary Of The Review:**

The main contribution, a method of combination of Uni-modal Ensemble and Uni-modal Teacher, seems practically utilizable and showing better performance for multi-modal joint training on deep neural networks.
Also, I think it is interesting idea to leverage distillation for generalized setting of multi-modal joint training.
However, the writing is confusing and not enough clear to be reproducible.

---

> ### Author Response · Authors · 2022-11-05
> **Thanks and response to concerns (Part 1)**
>
> Thanks for your review, below we address your concerns.
>
> **1, About "Modality Laziness"**
>
> > _**1.1** what is the exact definition of modality laziness_
>
> The definition of Modality Laziness is: the phenomenon of learning insufficient uni-modal representations of each modality in multi-modal joint learning.
>
> It first appears at the beginning of the third paragraph of the introduction chapter, where we not only give its definition but also illustrate Modality Laziness in Figure 1 to help readers understand it better.
>
> > _**1.2** Why should we utilize new term? What are the discriminating points for it?_
>
> Because fitting the training set does not require the model to learn all features, features of some modalities may be under-learned.
> It looks like some modalities are lazy, so we named this phenomenon Modality Laziness.
>
> Other than that, “Laziness” of "Modality Laziness" has a pejorative connotation, implying that the phenomenon is terrible and we need to fix it.
>
> This phenomenon is one of the core motivations of our paper, which is mentioned many times in the later sections of our paper,  so giving it an appropriate name is essential for the writing of the paper. This avoids the need to re-describe what this phenomenon is every time we mention this phenomenon.
>
>
> **2, About algorithm and reproducibility**
>
> > _**2.1** lack of explanation for core parts such as model architectures of uni-modal teacher?_
>
> _If the "model architectures" you mentioned here mean the backbone networks used in UMT_:
>
> Algorithm 1 is used to introduce the framework of UMT, and the network backbones can be various. So we didn't specify which network backbone to use in sec 3.3.
> The network architectures are directly related to the multi-modal tasks.
>
> For example, in UCF101 and ModelNet40, we use ResNet18 as the model architecture; in VGG-Sound and Kinetics-400, we use 3D ResNet as the backbone; When comparing UMT with AVSlowFast, we use SlowFast Networks as the RGB backbone. We introduce these in the training settings and the experiments subsection of UMT in Section 4.
>
> _If the "model architectures" you mentioned here mean the architecture of UMT_, Algorithm 1,Figure 4, sec 3.3 and Appendix A.4 are enough to illustrate that.
>
> > _**2.2** lack of explanation for ... loss functions with λ_task, λ_distill, λ_distill in Algorithm 1?_
>
> The task loss function is **also** directly related to the multi-modal tasks, so we didn't specify which task loss function to use in sec 3.3 , too.
> For example, in the classification task, the loss function is Cross-Entropy loss, while in the segmentation task in Appendix A.9, the task loss is pixel-level cross-entropy loss. λ_task is set as 1.
>
> The distillation loss function is MSELoss, and λ_distill is set as 50 for all the experiments of Section 4, which are introduced in Sec 4.3.
>
> > _**2.3** the citation of reference works, which also needs for reproducibility_
>
> As for reference works of UMT, we think the most relevant papers are those about knowledge distillation, and we have cited them in Sec 2.
>
> If we have missed relevant articles, please point them out and we will update the related work section.
>
> > **_**2.4** reproducibility_**
>
> For reproducibility, we not only give a detailed description of UME and UMT in sec3.3 and Appendix A.4, **but also provide the reproducibility code in Supplementary Material, which is visible to everyone**. We think it's enough for reproducing the experimental results.
>
> **After clarifying the above questions**, we think "The proposed solution is simple, and easy to reproduce." as Reviewer DNPE said.
>
> And we think that our paper is "easy to follow", as Reviewer AaMz said, and "The writing is fine, and the description is clear.", as Reviewer fVP6 said.
>
>
> **3, About Section 3.4**
>
> We argue that Sec 3.4 is closely related to context.
>
> (1) In sec 3.1, we empirically show that Modality Lazinesss exists in recent multi-modal late-fusion methods, while in sec 3.4, we give a theoretical analysis of that phenomenon from a feature learning perspective. **In sec 3.4, we repeatedly mentioned Modality Laziness, which is closely related to sec 3.1**
>
> (2) In sec 3.4, we prove that Modality Laziness does hurt the generalization ability of the multi-modal model in Theorem 3.4, which **provides a theoretical basis for the motivation of our paper**. As we said in the Introduction section, we "prove that it does hurt the
> the generalization ability of the model"
>
> (3) In sec 3.3, we offer guidance for multi-modal learning and introduce UME and UMT.  In sec3.4, we **justify UME by theorem 3.4**: ''Performance Laziness compares the performance of multimodal joint training approaches with Uni-Modal Ensemble, demonstrating that when uni-modal features dominate, combining uni-modal predictions is more effective''
> and **justify UMT by theorem 3.5**: ''We next prove that UMT indeed helps uni-modal feature learning and can also learn some easy-to-learn paired features in Theorem 3.5''.

---

> > ### Author Response · Authors · 2022-11-05
> > **Thanks and response to concerns (Part 2)**
> >
> > > _**3.1**,  Are uni-modal features and paired features also redefined in Section 3.4?_
> >
> > Uni-modal features are still a type of multi-modal feature that can be learned from uni-modal training, and paired features are still a type of multi-modal feature that can only be learned from cross-modal interaction.
> >
> > In Definition 3.1 and 3.2, **we did not redefine uni-modal features and paired features**, but mathematically define how they are generated. Without defining their generation process, we cannot make the following theoretical derivations.
> >
> > > _**3.2** Is there no experimental result related to Section 3.4?_
> >
> > (1) The experimental results of sec3.1 motivate almost everything in this article.
> > Sec 3.4 gives the Modality Laziness of sec3.1 a theoretical analysis.
> >
> > (2) Also, Sec 3.4 justify our method theoretically and the experiments in Sec 4 give experimental support.
> >
> >
> > > _**3.3**, It seems that theoretical part (Section 3.4) and the others are totally separated_
> >
> > **Based on the above analysis, we argue that Sec 3.4 is closely related to context.**
> >
> >
> > **4, compare with Modality Competition[1]**
> >
> > Thank you for providing this recent related work.
> >
> > Compared our paper with Modality Competition, we think there are two main differences.
> >
> > The first and most important difference is that their paper only explains why the failure occurs but did not give a working solution, while our solution works well.
> >
> > The second difference is that our analysis is from the feature learning perspective and focuses on modeling the learned and unlearned features. We also abstract multi-modal features into uni-modal and paired features, which helps us understand multi-modal learning. Our paper and Modality Competition explain the insufficient learning of features from different perspectives.
> >
> > [1] Modality Competition: What Makes Joint Training of Multi-modal Network Fail in Deep Learning? (Provably), ICML 2022

---

> > > ### Author Response · Authors · 2022-11-07
> > > **Rebuttal Revision**
> > >
> > > Thanks for your review again, and we have updated our main paper to clarify the confusing part:
> > >
> > > **1**, We have updated the part about UMT in Sec 3.3 and pointed out that the model architecture and loss function that we use for different datasets can be found in Sec 4.
> > >
> > > **2**, As for sec 3.4, we have pointed out its relationship to sec 3.1 and sec 3.3 at the beginning of section 3.4.
> > > It is also pointed out at the beginning of sec4 that we will experimentally verify the method that sec3.4 theoretically justifies in sec 4. We also **highlight** the connection between sec 3.4 and other (sub)sections.
> > >
> > > **3**, We compare our paper with Modality Competition[1] in sec 2.
> > >
> > > **All modified texts are marked in red**
> > >
> > > If you have any other questions, we are happy to provide additional discussion to allay your concerns.
> > >
> > > [1] Modality Competition: What Makes Joint Training of Multi-modal Network Fail in Deep Learning? (Provably), ICML 2022

---

> > > > ### Author Response · Authors · 2022-11-15
> > > > **Follow-up**
> > > >
> > > > Thank you again for your constructive comments and suggestions. If we have successfully addressed your questions, we would strongly appreciate an increased score. Otherwise, please let us know and we are happy to provide additional experiments and/or discussion to allay your concerns.

---

> > > > > ### Author Response · Authors · 2022-12-01
> > > > > **Do our responses address your concerns?**
> > > > >
> > > > > Dear Reviewer 5mXz:
> > > > >
> > > > > Thank you again for your constructive comments and suggestions.
> > > > >
> > > > > If we have successfully addressed your questions, we would strongly appreciate an increased score.
> > > > > Otherwise, please let us know and we are happy to provide additional experiments and/or discussion to allay your concerns.
> > > > >
> > > > > Regards,
> > > > >
> > > > > The authors.

---

> ### Author Response · Authors · 2022-12-07
> **Did our answer address your concerns?**
>
> Dear Reviewer:
>
> Thank you for taking the time to review our paper and providing constructive comments.
>
> We have provided very detailed answers to your concerns and updated the paper.
> It has been almost a month since we replied to you, and we sincerely want to know whether our reply has successfully addressed your concerns.
>
> If we address your concerns, we would strongly appreciate an increased score. Otherwise, please let us know and we are happy to provide additional experiments and/or discussion to allay your concerns.
>
> Regards,
>
> The authors.

---

### Decision · Program_Chairs · 2023-01-20

**Decision:**

Reject

**Justification For Why Not Higher Score:**

The paper's readability needs to be improved, even after the updates. The authors introduce many terms that are unclear and not well defined, e.g.:
  - "modality laziness" could for example be formally defined as linear-probing-accuracy(one-arm-of-multimodal-encoder) < linear-probing-accuracy(corresponding-unimodal-encoder), rather than simply saying "insufficient representation"
  - similarly "naive multimodal training" -- what type of model do the authors mean here: early fusion? late fusion? other?
  - it is not immediately clear that "features" are learned representations in the authors' parlance, rather than inputs, as a reader could intuitively assume. it becomes clear eventually that these 'features' are learned, but this is an unnecessary barrier on reading comprehension
  - it would be helpful to provide explanations provided to the reviewers (e.g. about the practicality of the method) also to the readers of the paper, and formulate them more clearly. e.g. in a discussion about the "practicality" of the method, focus on the number of hyper parameters that are needed and how difficult it may be to estimate them reliably, no need to repeat that the proposed method outperforms others, but rather explain why that may be the case
Additional experimental comparisons as suggested by the reviewers could significantly strengthen the paper and lift it over the bar. Also, the core idea of mono-modal distillation to improve multi-modal learning has been published before, in Multimodal Learning with Incomplete Modalities by Knowledge Distillation; Qi Wang, Liang Zhan, Paul Thompson, Jiayu Zhou; KDD 2020. The use of this approach to improve performance even when modalities are not missing remains novel, but this work should be cited.


**Justification For Why Not Lower Score:**

n/a

**Metareview: Summary, Strengths And Weaknesses:**

Summary

The paper investigates a problem in supervised deep neural network training, where sometimes a joint multi-modal (e.g. audio-visual) model fails to outperform well-trained uni-modal baselines followed by late fusion, despite having access to more information from cross-modal interactions. The paper makes two contributions: (1) authors provide a theoretical analysis of characteristics among features derived from uni-modal features inputs, and "paired" features, which can only be learned from cross-modal interaction. Authors find "modality laziness" when one modality dominates and prevents multimodal learning. This leads authors to propose (2) an improved methodology consisting of either a Uni-Modal Teacher (UMT) to distill from or just using simple averaging predictions of unimodal models (Uni-Modal Ensemble, UME), together with decision heuristics. Authors provide experimental validation on sound classification (VGG-Sound), audio-video data for action recognition (UCF-101, Kinetics-400), and front-rear view of object data (ModelNet40), outperforming baseline methods.

Strengths

- Multimodal learning is an important area, and the authors make a theoretical and empirical contribution to advance the field
- Authors provide detailed experimental results and ablations to support their main arguments
- The proposed method is quite straightforward and yet results are competitive
- Experiments should be largely reproducible and the paper's approach is principled and motivated.

Weaknesses

- The presentation of the work can be improved in places, as reviewers point out. For example, some terms are not clearly defined, tables don't have sufficient captions (mAP? top-1 accuracy?), and some information for reproducibility is missing. While authors have made updates to the paper, it still feels like they could sometimes have made more thorough edits in order to better capture the spirit of the reviewers' comments.
- Related work is missing (as pointed out by reviewers), and should be added to the discussion and experimental comparison in more depth, not just cited (e.g. [Ngiam et al]); even after the update, information is split across multiple sections and appendix. I would for example find it more intuitive to move results on vgg (the main dataset used in this work) out of the appendix section on "different datasets" into the main text
- While the proposed method is simple and effective, a unified framework, agnostic to the unimodal priors and working well for both essential priors (including video classification and action recognition tasks) and meaningless priors (including visual question answer task), would arguably be a more significant contribution with more novelty and more impact. The proposed approach ultimately relies on heuristics.


**Summary Of Ac-Reviewer Meeting:**

n/a